# D-CORE: INCENTIVIZING TASK DECOMPOSITION IN LARGE REASONING MODELS FOR COMPLEX TOOL USE

## ABSTRACT

Effective tool use and reasoning are essential capabilities for large reasoning models (LRMs) to address complex real-world problems. Through empirical analysis, we identify a prevalent "Lazy Reasoning" phenomenon, where LRMs frequently engage in repetitive and meaningless reflective reasoning. This occurs primarily due to their inadequate ability to decompose tasks when reasoning in complex tool use scenarios. To address this, we propose a two-stage training framework D-CORE (**D**ecomposing tasks and **Co**mposing **Re**asoning processes) that first incentivize the LRMs' task decomposition reasoning capability via self-distillation, followed by diversity-aware reinforcement learning (RL) to restore LRMs' reflective reasoning capability. D-CORE achieves robust tool-use improvements across diverse benchmarks and model scales. Experiments on BFCLv3 demonstrate superiority of our method: D-CORE-8B reaches 77.7% accuracy, surpassing the best-performing 8B model by 5.7%. Meanwhile, D-CORE-14B establishes a new state-of-the-art at 79.3%, outperforming 70B models despite being $5\times$ smaller.

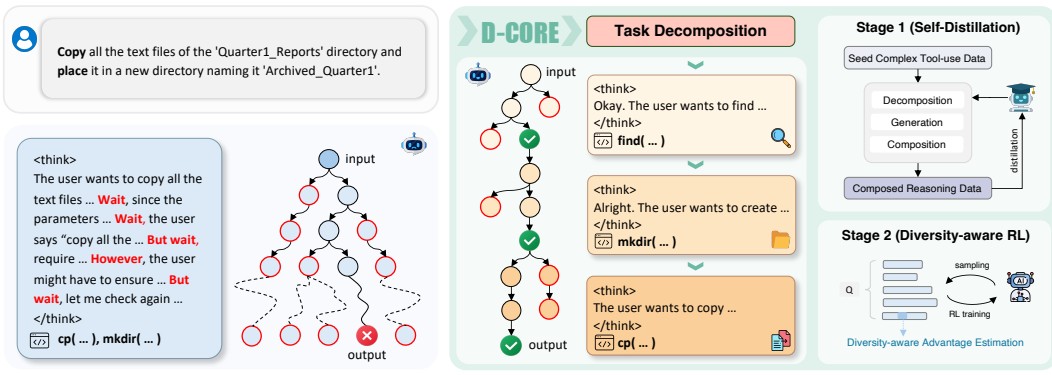

Figure 1: Comparison of baseline and D-CORE trained LRMs in complex tool use scenarios. Baseline LRMs exhibit "Lazy Reasoning" with repetitive reflection and incorrect answers, while D-CORE trained LRMs decompose tasks into executable subtasks.

## 1 INTRODUCTION

Tool use, also called function calling, refers to a large language model's (LLM's) ability to invoke external functions, APIs, or user-defined tools—an essential capability for agentic LLM applications. This functionality empowers LLMs to tackle increasingly complex and realistic tasks by enabling modular, structured, and interactive workflows (Qiao et al., 2024; Shen et al., 2024). As the field has advanced, benchmarks such as $\tau$-Bench (Yao et al., 2024) and BFCLv3 (Patil et al.) have highlighted the growing demand for robust agent reasoning and compositional tool use—capabilities that are essential for addressing real-world user needs.

Although various datasets (Liu et al., 2024b;a; Zhong et al., 2025; Prabhakar et al., 2025; Yin et al., 2025; Chen et al., 2023) and training methodologies (Lin et al., 2024; Qin et al., 2023; Chen et al.,

2024) have been proposed to support tool use capability, current approaches still rely on rule-based data construction and supervised fine-tuning(SFT), resulting in poor generalization and limited performance in complex scenarios (Chen et al., 2025; Chu et al., 2025). Meanwhile, the LRMs trained with RL combined with long-CoT reasoning have proven highly effective in mathematical and reasoning tasks (Guo et al., 2025; Yang et al., 2025; Anthropic, 2025; OpenAI, 2024b; 2025). While recent work demonstrates that RL-enhanced reasoning can be successfully applied to tool use (Qian et al., 2025; Zhang et al., 2025), LRMs face a critical bottleneck: they allocate significantly more tokens to reasoning in complex tool use scenarios, yet achieve negligible performance gains over LLMs. This disparity highlights a fundamental challenge: **how to effectively bridge the gap between reasoning capabilities and complex tool use**.

To address this question, we first evaluated the Qwen3 (Yang et al., 2025) series LRMs across diverse tool use tasks. While these LRMs demonstrate strong perceptual capabilities in parallel and tool-irrelevant scenarios, they suffer from "Lazy Reasoning" in complex multi-turn contexts—generating extensive yet meaningless reflective content and impeding reinforcement learning optimization (Yue et al., 2025; Gandhi et al., 2025; Ning et al., 2025). Our experiments trace this limitation to insufficient task decomposition capabilities and reasoning diversity (Patil et al.). Introducing decomposition-based prompting methods (Khot et al., 2022b; Zhou et al., 2022) effectively mitigates "Lazy Reasoning" behaviors, leading to significant improvements across multiple LRMs. Building on this insight, we propose D-CORE (**D**ecomposing tasks and **Co**mposing **Re**asoning processes), a comprehensive framework that systematically incentivizes both task decomposition reasoning capability and reasoning diversity in LRMs through self-distillation (Huang et al., 2022; Schick et al., 2023a; Madaan et al., 2023) and diversity-aware reinforcement learning (Wang et al., 2025; Cheng et al., 2025). Figure 1 illustrates how task decomposition converts inefficient reasoning cycles into effective step-by-step processes that yield correct answers.

In the self-distillation stage of D-CORE, we first sample seed complex tool use datasets and prompt LRMs to decompose queries into subtasks. We then feed these subtasks into LRMs to obtain reasoning processes and tool use results for each subtask, and finally organize the reasoning processes into trajectories based on different tool use scenario. Although self-distillation training effectively incentivizes the LRM's task decomposition reasoning capability, it simultaneously inhibits reflective behaviors—critical for complex problem-solving. To address this limitation, we propose a diversity-aware reinforcement learning that incorporates entropy-based bias terms into the advantage function, steering the LRM toward high-entropy reasoning paths enriched with reflective tokens. LRMs trained with our method can automatically decompose complex tasks into subtasks during its reasoning process and, based on its current progress, select appropriate tools to solve each subtask. Our main contributions are:

- We identify and formalize the "Lazy Reasoning" phenomenon—a fundamental limitation that affects LRMs in complex tool use scenarios—and provide both theoretical analysis and empirical validation of its impact on LRM performance.

- We develop a novel self-distillation framework that integrates task decomposition with reasoning process composition, enabling LRMs to acquire sophisticated sequential tool use strategies during reasoning without requiring additional human annotation.

- We propose a reinforcement learning framework that incorporates entropy-based advantage functions to enable self-distillation LRMs to restore their exploration and reflection capabilities while maintaining task decomposition abilities, thereby addressing more complex tool use scenarios.

## 2 TOOL USE REASONING: PATTERNS AND LIMITATIONS

### 2.1 PRELIMINARY

**Tool use tasks.** Tool use tasks can be categorized into single-turn and multi-turn tasks based on context dependency. Single-turn tasks can be formulated as Markov decision process $M(P, T, Q) \to \tau$, where $P$ denotes the system policy, $T$ represents the available tool set, $Q$ corresponds to the current query, and $\tau$ represents tool call results. Upon decomposing query $Q$ into subtasks $S = \{s_1, s_2, \ldots, s_n\}$,

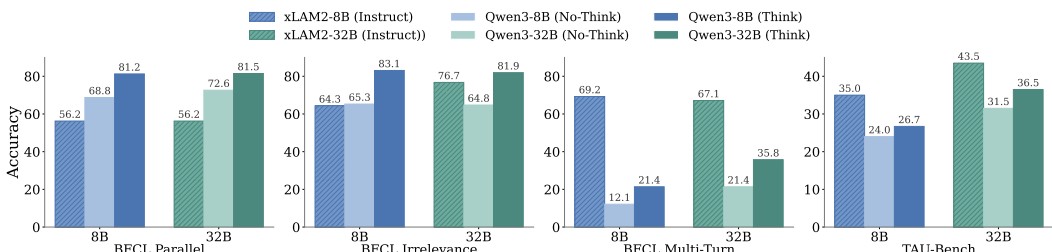

Figure 2: Comparison of LRM Qwen3 vs. instruct LLM xLAM2 performance on BFCLv3 Parallel, Irrelevance , Multi-turn and $\tau$-bench task.

three key scenarios for $M$ arise:

$$M \in \begin{cases} \text{Sequential: } s_i \text{ depends on output of } s_{i-1}, \\ \text{Parallel: } s_i \text{ can execute parrallely}, \\ \text{Irrelevant: } Q \text{ requires no tool use}. \end{cases}$$

The primary challenge lies in the fact that multi-intent and tool irrelevance $Q$ make $M$ particularly challenging for tool use LLMs (Liu et al., 2024a; Lin et al., 2024). Furthermore, multi-turn tool use scenarios (Yao et al., 2024; Patil et al.; Prabhakar et al., 2025) can be formulated as a $M(P, T, C, Q) \rightarrow \tau$, where $C$ denotes the conversation history. Unlike single-turn tasks, this formulation introduces additional complexity by requiring consideration of both the current query $Q$ intent and the long-term intent embedded in $C$.

**Reasoning process.** A reasoning process $\mathcal{R}$ refers to the sequence of intermediate steps through which a LRM arrives at its final answer. Within the LRM outputs examined in this paper, reasoning processes are specifically delimited by `<think>` and `</think>` tags. We controlled LRMs' reasoning processes through prompts containing `"<think>\n\n</think>\n\n"` tags to enable "no-think" modes. A thought $r$ is the basic logical block in a reasoning process. Given a reasoning process $\mathcal{R}$, it can be decomposed into an ordered sequence of thoughts: $\mathcal{R} = \{r_1, r_2, ..., r_n\}$, where $n$ represents the total number of thoughts.

## 2.2 REASONING PROCESS ENHANCES TOOL USE AWARENESS

To examine how reasoning process influences tool use performance, we conduct evaluations on parallel, tool irrelevance, multi-turn tasks from BFCLv3 (Patil et al.), as well as on $\tau$-bench (Yao et al., 2024). Our evaluation employs the LRM Qwen3 (Yang et al., 2025) series and tool-use-specialized instruct LLM xLAM2 (Prabhakar et al., 2025) series. For Qwen3, we perform additional experiments in "no-think" mode to isolate reasoning capabilities. As shown in Figure 2, vanilla LRMs outperform both "no-think" LRMs and instruct LLMs on parallel and irrelevance tasks, revealing that explicit reasoning effectively determines the structural relationship between query $Q$ and tool set $T$ by identifying tool requirements, dependencies, and relevance patterns. This finding reveals the tremendous potential of LRMs in the tool use tasks. However, in complex tool use scenarios such as BFCLv3 multi-turn and $\tau$-bench, LRMs still exhibit substantial performance gaps compared to specialized fine-tuned instruct LLMs, motivating our investigation into this limitation.

## 2.3 LAZY REASONING IN TOOL USE REASONING PROCESS

**Reasoning behavior variations in tool use.** To investigate why LRMs struggle in multi-turn scenarios, we sample 20 rollouts per question on BFCL v3 single-turn and multi-turn tasks using Qwen3-8B, with MATH-500 (Hendrycks et al., 2021) as a mathematical baseline for comparison with prior work (Ning et al., 2025; Bogdan et al., 2025; Venhoff et al., 2025). Each rollout's reasoning process is segmented into thoughts and classified into four behavior types: (1) Task Decomposition— breaking problems into steps, (2) Reflection—revising when errors occur, (3) Verification—checking results, and (4) Deduction—drawing conclusions from assumptions, following the approach in Ning et al. (2025), to provide a detailed analysis of the impact of reasoning processes on LRMs.

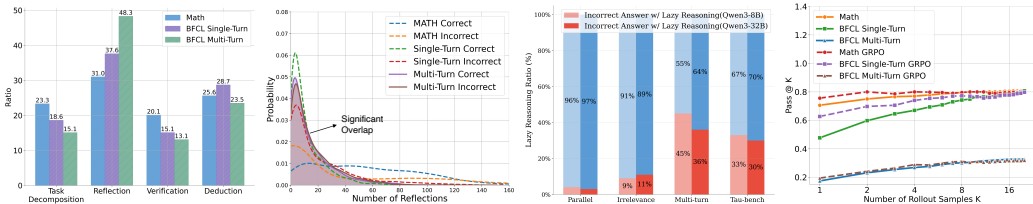

(a) Four Behaviors    (b) Lazy Reasoning    (c) Lazy Reasoning Ratio    (d) Reasoning Capability

Figure 3: **(a)** Distribution of four behavioral categories across rollout samples. **(b)** Probability density function of reflections on rollout samples.**(c)** Ratio of Lazy Reasoning in different tasks.**(d)** Pass@k performance metrics following GRPO.

Figure 3a reveals a striking difference: multi-turn scenarios reasoning processes contain significantly less task decomposition but substantially more reflection compared to single-turn and math scenarios reasoning processes. This suggests LRMs are "spinning their wheels" rather than systematically breaking down complex problems. Figure 3b provides compelling evidence: probability distributions for correct and incorrect answers nearly overlap in multi-turn tasks, unlike the distinct distributions in single-turn and math tasks. This overlap reveals that extensive low-information reflec-

> **Lazy Reasoning**
>
> Lazy Reasoning occurs during complex tool use tasks where LRMs replace effective task decomposition with excessive, low-information reflection, yielding long, unproductive reasoning with little improvement in answer accuracy.

tion make reasoning ineffective—a phenomenon we term **"Lazy Reasoning"**, defined as processes exceeding 300 tokens and 3 reflections. These thresholds come from histogram analysis and are adjustable per domain. As shown in Figure 3c, multi-turn failures are strongly correlated with Lazy Reasoning, suggesting that addressing this issue could significantly improve performance.Details are in Appendix A.1.

**Discussions on Lazy Reasoning.** Since reasoning capabilities are elicited during the RL process, one might ask whether more RL helps addressing Lazy Reasoning; our results show it does not—Lazy Reasoning persists and even impedes optimization. We applied GRPO (Shao et al., 2024) to Qwen3-8B using mathematics and tool use datasets. Figure 3d shows that Lazy Reasoning prevents RL from optimizing reasoning capability (Yue et al., 2025), resulting in minimal pass@1 improvement in multi-turn task after training. Further theoretical analysis is provided in the Appendix A.1.1.

**Addressing Lazy Reasoning.** Recent analysis (Patil et al.) reveals that LLMs fail in multi-turn tasks primarily due to poor understanding of subtask execution results and user queries, indicating deficient planning and execution capabilities for complex tasks— consistent with our observation of Lazy Reasoning. We hypothesize that Lazy Reasoning serves as **a compensatory mechanism adopted by LRMs due to insufficient task decomposition capabilities**—LRMs default to unproductive trial-and-error when unable to properly break down complex problems.

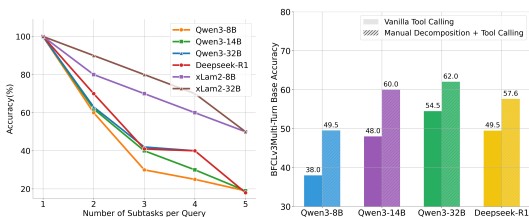

(a) Manual Composition    (b) Manual Decomposition

Figure 4: **(a)** Accuracy degradation caused by subtask composition. **(b)** Accuracy improvement through manual task decomposition.

To study the causes of Lazy Reasoning, we created multi-subtask queries by combining simple single-subtask queries (Chen et al., 2024). Figure 4a shows that LRMs performance significantly decrease as the number of subtasks increases, indicating that task decomposition complexity positively correlates with Lazy Reasoning. Prior works offer concrete mechanisms to endow LLMs with task decomposition capability: decomposition-based prompting methods such as Decomposed Prompting (Khot et al., 2022b) and Least-to-Most Prompting (Zhou et al., 2022) implement planning-and-execution by breaking a complex goal into subtasks and integrating intermediate results. We adopt these methods to manually supply reliable decompositions and enforce step-by-step execution (A.1.5). Results in Figure 4b confirms our hypothesis: appropriate task decomposition improved the effectiveness of reasoning, thereby significantly enhancing LRM performance on multi-turn tasks.

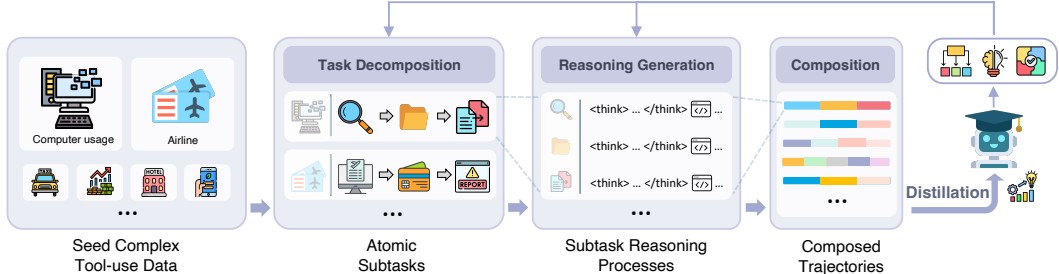

Figure 5: Overview of D-CORE self-distillation stage. Through self-distillation, a LRM with task decomposition and subtask execution capabilities is acquired.

## 3 D-CORE

Building on the effectiveness of decomposition-based prompting in enhancing the complex tool use capabilities of LRMs, we propose D-CORE—a method that enables LRMs to autonomously learn tool use in complex scenarios through two stages: self-distillation for incentivize task decomposition and subtask execution capabilities during reasoning, and diversity-aware GRPO to mitigate gradient vanishing during RL and strengthen reflective reasoning.

### 3.1 INCENTIVIZE COMPLEX TASK DECOMPOSITION REASONING VIA SELF-DISTILLATION.

Task decomposition enhances tool use capabilities, but injecting this ability into reasoning typically requires a powerful teacher LRM to generate reasoning trajectories with decomposition steps. We take a different view: *current LRMs can generate high-quality reasoning trajectories on their own when given explicit structural guidance.* Our self-distillation framework, inspired by self-improvement work (Huang et al., 2022; Schick et al., 2023a; Madaan et al., 2023), provides this guidance without external supervision.

**Task Decomposition.** We sample seed datasets and prompt LRM $\mathcal{M}$ to decompose queries $Q$ into subtasks $\mathcal{S} \leftarrow \text{Decompose}(\mathcal{C}, Q, Y^*, \mathcal{M})$, where $\mathcal{C} = \{P, T, C\}$ represents comprehensive contextual information: system policy $P$, tool set $T$, conversation history $C$. The reference trajectories $Y^*$ are provided along with few-shot examples to improve decomposition success rates, serving as the most critical factor for enhancing task decomposition performance. The complete prompt can be found in Appendix A.2.

**Reasoning Generation.** Given the decomposed subtasks $\mathcal{S} = \{s_1, s_2, \ldots, s_n\}$, we feed them into LRM $\mathcal{M}$ to generate reasoning processes and tool calls $(\mathcal{R}_i, \tau_i) \leftarrow \mathcal{M}(\mathcal{C}, s_i)$. The generation process varies by scenario: sequential subtasks are processed iteratively to maintain context dependency, while parallel subtasks are handled simultaneously. For tool-irrelevant queries where decomposition are not applicable, we prompt $\mathcal{M}$ to generate explainations for why task cannot be decomposed.

**Composition.** At composition stage, for sequential and parallel scenario, we compose subtasks $\mathcal{S}$, reasoning process $\mathcal{R}$, tool calls $\tau$ and tool response $o$ into complete reasoning trajectories $\hat{\mathcal{Y}} \leftarrow \text{Compose}(\{(s_i, \mathcal{R}_i, \tau_i, o_i)\}_{i=1}^{|\mathcal{S}|})$ while $\hat{\mathcal{Y}} \leftarrow \text{Compose}(\mathcal{R}, Y^*)$ for tool-irrelevant scenario. Reflection mechanisms are incorporated in Compose template for parallel and irrelevant scenarios. See Appendix A.3 for details.

**Distillation.** Based on the composed reasoning trajectories $\hat{\mathcal{Y}}$, we apply SFT on LRM $\mathcal{M}$. During the SFT, LRM acquire task decomposition and subtask execution capabilities through parameters $\theta$ by maximizing the probability $\pi_\theta(\hat{\mathcal{Y}}_t \mid (\mathcal{C}, Q), \hat{\mathcal{Y}}_{1:t-1})$, the optimization loss function can be expressed as follows:

$$\mathcal{L}_{\text{self-distillation}}(\theta) = -\mathbb{E}[\log \pi_\theta(\hat{\mathcal{Y}}_t \mid (\mathcal{C}, Q), \hat{\mathcal{Y}}_{1:t-1})]. \tag{1}$$

Within our self-distillation framework, LRM $\mathcal{M}$ produces appropriate subtasks and associated reasoning processes for each, creating a comprehensive training dataset that captures the LRM's own problem-solving methodology while addressing its inherent limitations.

---

**Algorithm 1** Task Decomposition and Reasoning Composition

---

**Input:** Context $\mathcal{C} = \{P, T, C\}$ where $P$ is system policy, $T$ is tool set, $C$ is conversation history, query $Q$, LRM $\mathcal{M}$, reference trajectories $Y^*$
**Output:** Composed reasoning trajectories $\hat{\mathcal{Y}}$
1: $\mathcal{S} \leftarrow \text{Decompose}(\mathcal{C}, Q, Y^*, \mathcal{M})$           ▷ *Decompose task into subtasks*
2: **return** $\emptyset$ **if** $|\mathcal{S}| \neq |Y^*|$        ▷ *Verify that subtask and reference trajectories counts match.*
3: **if** $\mathcal{S}$ is SEQUENTIAL **then**          ▷ *Reasoning Generation and Composition*
4:      $\mathcal{I}_0 \leftarrow \text{InitInput}(\mathcal{C})$
5:      **for** $i = 1, \ldots, |\mathcal{S}|$ **do**
6:          $(\mathcal{R}_i, \tau_i) \leftarrow \mathcal{M}(\mathcal{I}_{i-1}, s_i)$          ▷ *Reasoning & tool selection*
7:          $o_i \leftarrow \text{Execute}(\tau_i)$          ▷ *Tool execution*
8:          $\mathcal{I}_i \leftarrow \text{Update}(\mathcal{I}_{i-1}, \tau_i, o_i)$
9:      **end for**
10:     $\hat{\mathcal{Y}} \leftarrow \text{Compose}_{\text{seq}}(\{(s_i, \mathcal{R}_i, \tau_i, o_i)\}_{i=1}^{|\mathcal{S}|})$
11: **else if** $\mathcal{S}$ is PARALLEL **then**
12:      **for** $i = 1, \ldots, |\mathcal{S}|$ **do**
13:          $(\mathcal{R}_i, \tau_i) \leftarrow \mathcal{M}(\mathcal{C}, s_i)$          ▷ *Parallel processing*
14:      **end for**
15:     $\hat{\mathcal{Y}} \leftarrow \text{Compose}_{\text{par}}(\{(s_i, \mathcal{R}_i, \tau_i)\}_{i=1}^{|\mathcal{S}|})$
16: **else**          ▷ *Tool-irrelevant*
17:      $\mathcal{R} \leftarrow \mathcal{M}(\mathcal{C}, Q)$
18:      $\hat{\mathcal{Y}} \leftarrow \text{Compose}_{\text{irr}}(\mathcal{R}, Y^*)$
19: **end if**
20: **return** $\hat{\mathcal{Y}}$ **if** $\text{Verify}(\hat{\mathcal{Y}}, Y^*)$ **else** $\emptyset$          ▷ *Verify whether $\hat{\mathcal{Y}}$ are consistent with the $Y^*$*

---

## 3.2 DIVERSITY-AWARE REINFORCEMENT LEARNING

While self-distillation equips LRMs with task decomposition capabilities, it significantly reduces reflective reasoning—a critical component for exploration and robust performance. This diversity collapse is clearly reflected in the reward standard deviation of LRMs' rollouts. As shown in Figure 6a, self-distilled model exhibits near-zero reward std. To understand why this poses a problem for GRPO, recall that the advantage $A_{i,t}$ is defined as:

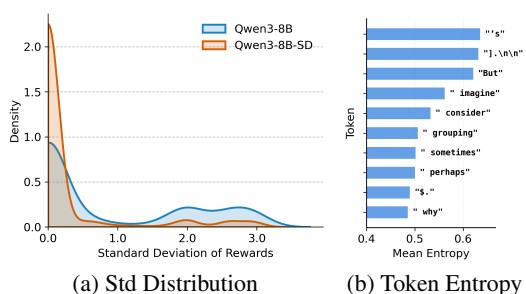

(a) Std Distribution      (b) Token Entropy

$$A_{i,t} = \frac{R_i - \text{mean}(\{R_i\}_{i=1}^G)}{\text{std}(\{R_i\}_{i=1}^G)}, \quad (2)$$

Figure 6: **(a)** Effect of self-distillation on reward variance. SD: self-distillation. **(b)** Top 10 high-entropy tokens for Qwen3-8B-SD.

where $G$ is the rollouts number. For simplicity, we omit the KL penalty term, and the objective function of GRPO is formulated as:

$$\mathcal{J}_{\text{GRPO}}(\theta) = \mathbb{E}[\min(\frac{\pi_\theta(y_{i,t} \mid x_i, y_{i,<t})}{\pi_{\text{ref}}(y_{i,t} \mid x_i, y_{i,<t})} A_{i,t}, \text{clip}(\frac{\pi_\theta(y_{i,t} \mid x_i, y_{i,<t})}{\pi_{\text{ref}}(y_{i,t} \mid x_i, y_{i,<t})}, 1 - \epsilon, 1 + \epsilon)A_{i,t})]. \quad (3)$$

For clarity, we omit the clipping term in subsequent derivations. The resulting policy gradient of GRPO can be expressed as:

$$\nabla_\theta \mathcal{J}_{\text{GRPO}} = \frac{1}{G}\left(\sum_{(i,t)} \frac{1}{|y_i|} \frac{\pi_\theta(y_{i,t} \mid x_i, y_{i,<t})}{\pi_{\text{ref}}(y_{i,t} \mid x_i, y_{i,<t})} A_{i,t} \nabla_\theta \log \pi_\theta(y_{i,t}|x_i, y_{i,<t})\right), \quad (4)$$

when std approaches zero, the advantage $A_{i,t}$ vanishes and eliminates the gradient. This motivates us to seek an alternative formulation. Recent work (Wang et al., 2025; Cheng et al., 2025) shows that high-entropy tokens correlate with reflective reasoning, which we confirm in Figure 6b, and that entropy-based advantages promote reasoning diversity (Cheng et al., 2025). This leads to our question: *can we leverage an entropy-based advantage to address gradient vanishing when std approaches zero, thus recovering reasoning diversity?*

We propose Diversity-Aware GRPO (DA-GRPO) with entropy-based advantage shaping mechanism that incentivizes high-entropy reasoning. The advantage in DA-GRPO can be formulated as:

$$\hat{A}_{i,t} = \begin{cases} \psi(\mathcal{H}_{i,t}) & \text{if } A_{i,t} < \zeta, \\ A_{i,t} & \text{otherwise,} \end{cases} \tag{5}$$

where $\zeta$ is a small constant for numerical stability (e.g., $\zeta = 10^{-8}$) and $\psi(\mathcal{H}_{i,t})$ is an entropy-based advantage term:

$$\psi(\mathcal{H}_{i,t}) = \min(\alpha \cdot \mathcal{H}_{i,t}^{\text{detach}}, \delta), \tag{6}$$

in which larger $\alpha$ promotes high-entropy token generation, while $\delta$ bounds the entropy advantage. The entropy term $\mathcal{H}_{i,t}^{\text{detach}}$ is detached during backpropagation, serving as a fixed offset that adjust update magnitude without affecting gradients. It is computed over the token vocabulary $\mathcal{V}$ as:

$$\mathcal{H}_{i,t}^{\text{detach}} = -\sum_{v \in \mathcal{V}} \pi_\theta(v|x_i, y_{i,<t}) \log \pi_\theta(v|x_i, y_{i,<t}). \tag{7}$$

The key idea is to scale each token's gradient update in proportion to its detached entropy $\mathcal{H}_{i,t}^{\text{detach}}$ when std approaches zero. The objective function of DA-GRPO is formulated as:

$$\mathcal{J}_{\text{DA-GRPO}}(\theta) = \mathbb{E}[\min(\frac{\pi_\theta(y_{i,t} \mid x_i, y_{i,<t})}{\pi_{\text{ref}}(y_{i,t} \mid x_i, y_{i,<t})}\hat{A}_{i,t}, \text{clip}(\frac{\pi_\theta(y_{i,t} \mid x_i, y_{i,<t})}{\pi_{\text{ref}}(y_{i,t} \mid x_i, y_{i,<t})}, 1-\epsilon, 1+\epsilon)\hat{A}_{i,t})$$
$$- \lambda \mathbb{D}_{\text{KL}}[\pi_\theta || \pi_{\text{ref}}]. \tag{8}$$

The reward employed in DA-GRPO aligns with that of ToolRL (Qian et al., 2025)(Appendix A.8):

$$R_i = R_{\text{format}} + R_{\text{struct}} + R_{\text{key}} + R_{\text{value}}. \tag{9}$$

Appendix A.9 provides a rigorous proof that DA-GRPO exhibits inherently stable and self-limiting properties. As a diversity-aware mechanism, it effectively addresses the optimization difficulties induced by self-distillation while restoring reasoning diversity in LRMs.

## 4 EXPERIMENTS

### 4.1 EXPERIMENTAL SETUPS

We collect data from two sources: subsets of open-source datasets (Liu et al., 2024a;b; Prabhakar et al., 2025) and trajectories of our custom-built tool use agent, ensuring coverage across diverse domains and difficulty levels. From these sources, we generate 40,000 self-distillation instances and sample 5,000 instances for RL. We use Qwen3-8B/14B (Yang et al., 2025) as the backbone LRM, with packing technique (Xu et al., 2024) for self-distillation and verl framework (Sheng et al., 2024) for RL training. Details are provided in Appendix A.6. We evaluate on two realistic agent benchmarks: BFCLv3 (Patil et al.) and $\tau$-bench (Yao et al., 2024). We compare against open-source models (Guo et al., 2025; Liu et al., 2024a; Prabhakar et al., 2025), as well as closed-source LRMs and LLMs (OpenAI, 2024b; Anthropic, 2025; DeepMind, 2024; OpenAI, 2024a).

### 4.2 MAIN RESULTS

| Model | $\tau$-retail (%) | $\tau$-airline (%) | Avg (%) |
|---|---|---|---|
| Reported Results | | | |
| Llama3.1-70B-Instruct | 50.4 | 26.0 | 38.2 |
| Deepseek-R1 | 55.6 | 40.0 | 47.8 |
| GPT-4o | 62.8 | 43.0 | 52.9 |
| Claude 3.7 Sonnet | **78.3** | 41.2 | 59.8 |
| o1 | 73.5 | **54.2** | 63.9 |
| Reproduced Results | | | |
| Qwen3-8B | 34.7 | 23.2 | 29.0 |
| Qwen3-14B | 41.6 | 25.6 | 33.6 |
| Qwen3-32B | 39.6 | 33.6 | 36.6 |
| xLAM2-8B | 50.7 | 34.0 | 42.4 |
| xLAM2-32B | 52.2 | 37.6 | 45.0 |
| xLAM2-70B | **57.7** | 38.8 | 48.3 |
| ToolRL-Qwen3-8B | 40.6 | 22.4 | 31.5 |
| ToolRL-Qwen3-14B | 49.5 | 26.0 | 37.8 |
| D-CORE-8B(ours) | 50.7 | 44.4 | 47.6 |
| $\triangle$ from Qwen3-8B | +16.0 | +21.8 | +18.6 |
| D-CORE-14B(ours) | 56.5 | **46.0** | **51.3** |
| $\triangle$ from Qwen3-14B | +14.9 | +20.4 | +17.7 |

Table 1: Performance comparison on $\tau$-bench. The best results are bolded. Reproduced results are averaged over 5 independent runs.

$\tau$-bench. As shown in Table 1, D-CORE achieves substantial gains on $\tau$-bench, improving accuracy by 18.6% for Qwen3-8B and 17.7% for Qwen3-14B, validating the effectiveness and scalability of our approach. Notably, D-CORE-14B excels in the $\tau$-airline task with the highest accuracy of

Table 2: Comparison of the accuracy on BFCLv3 Single-Turn, Hallucination measurement and Multi-Turn tasks. The best results are bolded.

| Model | BFCLv3 | | | | | | | | | |
|---|---|---|---|---|---|---|---|---|---|---|
| | Overall | Single-Turn | | Hallucination | | Multi-Turn | | | | |
| | Accuracy (%) | Live | Non-Live | Irrelevance | Relevance | Base | Miss Func | Miss Param | Long Context | Overall (%) |
| Proprietary Model Results | | | | | | | | | | |
| Claude-3.7-Sonnet | 58.6 | 78.4 | 41.3 | 81.4 | 72.2 | 54.5 | 45.0 | 45.5 | 48.5 | 48.4 |
| Gemini-2.5-Pro | 59.9 | 74.6 | 65.3 | **90.7** | 33.3 | 39.5 | 29.5 | 31.5 | 38.0 | 34.6 |
| DeepSeek-R1 | 63.8 | 77.3 | 75.5 | 73.3 | 77.8 | 49.5 | 31.5 | 32.0 | 42.5 | 38.9 |
| o1 | 67.8 | 80.6 | 85.7 | 87.8 | 72.2 | 51.0 | 0.5 | 48.5 | 44.5 | 36.1 |
| GPT-4.5 | 70.3 | 79.3 | 86.1 | 83.6 | 66.7 | 58.0 | 37.5 | 43.0 | 43.0 | 45.4 |
| GPT-4o | 71.7 | 78.8 | 86.8 | 81.3 | 83.3 | 61.0 | 45.5 | 35.5 | 58.0 | 50.0 |
| Open-Source Model Results | | | | | | | | | | |
| Qwen3-8B | 66.3 | 78.5 | 88.8 | 79.1 | 77.8 | 38.5 | 32.0 | 31.5 | 30.0 | 33.0 |
| Qwen3-14B | 65.9 | 80.0 | 88.2 | 81.0 | 72.2 | 48.0 | 33.0 | 33.5 | 32.0 | 36.6 |
| ToolACE-2-8B | 68.7 | 80.1 | 87.6 | 90.1 | 72.2 | 49.0 | 29.0 | 28.0 | 42.0 | 37.0 |
| Qwen3-32B | 69.2 | 77.3 | 88.9 | 75.8 | 72.2 | 54.5 | 41.0 | 30.5 | 46.5 | 43.1 |
| xLAM2-8B | 72.0 | 66.7 | 84.4 | 64.3 | 77.8 | 75.0 | 71.0 | 64.5 | 66.0 | 69.1 |
| xLAM2-32B | 76.4 | 74.2 | 89.3 | 76.7 | **88.9** | **79.0** | 71.5 | 63.5 | 54.5 | 67.1 |
| xLAM2-70B | 78.4 | 72.9 | 88.4 | 78.9 | 66.7 | 77.5 | **75.5** | **73.0** | **74.0** | **75.0** |
| Custom-Trained Model Results | | | | | | | | | | |
| ToolRL-Qwen3-8B | 65.9 | 82.4 | 88.9 | 85.9 | 77.8 | 30.5 | 35.5 | 21.0 | 20.0 | 26.8 |
| ToolRL-Qwen3-14B | 68.5 | 81.5 | **89.4** | 81.9 | 72.2 | 41.0 | 41.0 | 31.5 | 29.5 | 35.8 |
| D-CORE-8B(ours) | 77.7 | 82.2 | 86.4 | 88.2 | 77.8 | 75.0 | 68.0 | 59.0 | 53.0 | 63.8 |
| *Δ from Qwen3-8B* | +11.4 | +3.7 | -2.4 | +9.1 | 0.0 | +36.5 | +36.0 | +27.5 | +23.0 | +30.8 |
| D-CORE-14B(ours) | **79.3** | **82.9** | 87.2 | 89.2 | 72.2 | 78.0 | 66.0 | 64.0 | 61.5 | 67.4 |
| *Δ from Qwen3-14B* | +13.4 | +2.9 | -1.0 | +8.4 | 0.0 | +30.0 | +33.0 | +30.5 | +29.5 | +30.8 |

46.0%, where the LRM handles complex refund evaluation and compensation decisions, requiring 4-5 subtasks per query when user intentions are unclear.

**BFCLv3.** Table 2 reveals that D-CORE achieves substantial accuracy gains of 11.4% and 13.4% for Qwen3-8B and Qwen3-14B respectively, with remarkable improvements of 30.8% on challenging multi-turn tasks. In contrast, the baseling ToolRL method using GRPO alone fails to deliver meaningful improvements in multi-turn scenarios, highlighting the effectiveness of our approach. D-CORE-8B establishing new state-of-the-art among 8B models at 77.7% overall accuracy, significantly outperforming xLAM2-8B. D-CORE-14B achieves 79.3% accuracy with 5× fewer parameters than the previous state-of-the-art 70B LLM, validating LRM's test-time scaling effectiveness.

## 4.3 ABLATION STUDY AND ANALYSIS

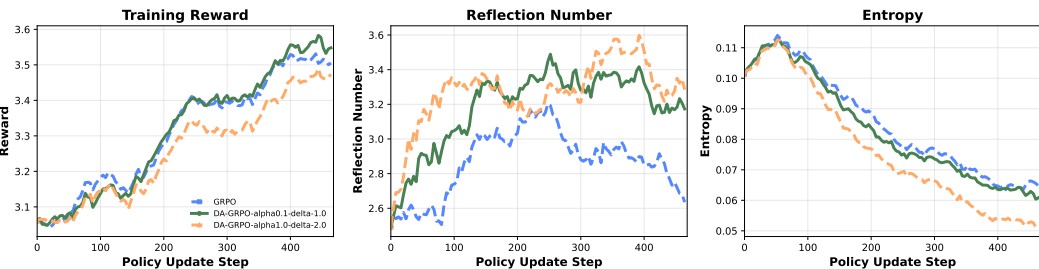

Figure 7: Training dynamics comparison across GRPO and DA-GRPO on D-CORE-8B.

**Training dynamics.** Figure 7 shows the training dynamics of D-CORE-8B. With $\alpha$=0.1, DA-GRPO achieves the highest reward while consistently increasing reflection tokens. At $\alpha$=1.0, reflection tokens increase progressively but rewards remain lower than GRPO. This revealing a trade-off: entropy-based advantages encourage reflection, but excessive $\alpha$ causes exploration collapse.

**Effectiveness of Self-Distillation.** As shown in Table 5, self-distillation demonstrates improved effectiveness with increasing sample size and mitigates RL optimization challenges. Notably, when

Table 3: Accuracy of BFCLv3 and $\tau$-Bench, where SD represents self-distillation.

| Model | Avg | BFCLv3 | | | | | $\tau$-Bench | |
| | Accuracy (%) | Single-Turn Live(%) | Single-Turn Non-Live(%) | Relevance (%) | Irrelevance (%) | Multi-Turn Overall(%) | Retail (%) | Airline (%) |
|---|---|---|---|---|---|---|---|---|
| Qwen3-8B | 38.1 | 78.5 | 88.8 | 77.8 | 79.1 | 33.0 | 34.7 | 23.2 |
| +SD | 46.2 | 82.5 | 86.5 | 66.7 | 89.7 | 57.5 | 42.0 | 31.2 |
| +GRPO | 55.6 | 75.6 | 85.4 | 72.2 | 84.5 | 67.4 | 49.2 | 41.2 |
| $\triangle$ from SD | +9.4 | -6.9 | -1.1 | +5.5 | -5.2 | +9.9 | +7.2 | +10.0 |
| +DA-GRPO$_{\alpha=0.01,\delta=0.5}$ | 56.2 | 77.0 | 86.7 | 77.8 | 83.7 | 67.9 | 52.5 | 38.8 |
| +DA-GRPO$_{\alpha=0.1,\delta=0.5}$ | 57.6 | 82.3 | 86.4 | 77.8 | 89.4 | 63.8 | 50.7 | 44.4 |
| +DA-GRPO$_{\alpha=0.4,\delta=0.5}$ | 55.5 | 78.8 | 86.1 | 72.2 | 85.5 | 59.9 | 54.6 | 36.8 |
| +DA-GRPO$_{\alpha=1.0,\delta=1.0}$ | 54.0 | 76.4 | 84.8 | 83.3 | 84.4 | 62.1 | 47.9 | 39.6 |
| $\triangle$ from SD | +11.4 | -0.2 | -0.1 | +11.1 | -0.3 | +6.3 | +8.7 | +13.2 |

Table 4: Accuracy on out-of-distribution tasks. The best results are bolded.

| Model | $\tau^2$-Bench | | | | ACEBench-en | | | | BFCLv4-Agentic | |
| | Retail | Airline | Telecom | Overall | Normal | Special | Agent | Overall | Web-base | Memory |
|---|---|---|---|---|---|---|---|---|---|---|
| Qwen3-8B | 41.5 | 31.3 | 26.3 | 33.0 | 71.4 | 75.3 | 29.1 | 65.9 | 16.0 | 18.9 |
| Qwen3-14B | 46.5 | 30.0 | 31.7 | 36.1 | 66.9 | 84.0 | 44.2 | 68.0 | 34.0 | 25.2 |
| Qwen3-32B | 49.1 | 36.0 | 28.4 | 37.8 | 75.9 | 77.3 | 49.2 | 72.2 | 32.0 | 25.0 |
| xLAM2-8B | 51.3 | 35.4 | 22.4 | 36.4 | 58.8 | 0.0 | 5.0 | 34.8 | 8.0 | 15.7 |
| xLAM2-32B | 53.1 | 41.6 | 26.1 | 40.3 | 69.2 | 24.7 | 13.4 | 52.5 | 31.0 | 16.8 |
| xLAM2-70B | 54.9 | 46.0 | 29.3 | 43.4 | 57.1 | 5.3 | 38.4 | 36.5 | 13.0 | 17.6 |
| D-CORE-8B (ours) | 49.8 | 38.4 | 30.5 | 39.6 | 77.9 | 78.7 | 59.2 | 75.2 | 36.0 | 20.4 |
| D-CORE-14B (ours) | 53.5 | 44.1 | 34.9 | 44.2 | 77.1 | 88.7 | 55.8 | 76.9 | 39.0 | 26.5 |

GRPO is applied to Qwen3-8B, improvements are marginal or even negative. In contrast, a self-distilled model achieves substantial gains of 9.9% and 8.6% on the respective benchmarks under identical settings, highlighting the efficacy of our approach.

**Effectiveness of DA-GRPO.** Figure 8a shows that self-distillation significantly increases task decomposition while reducing reflection in thoughts. DA-GRPO subsequently restores some reflection proportion. Table 3 demonstrates DA-GRPO ensures RL optimization targets both correct tool usage and increased reflective reasoning. However, as $\alpha$ increases, tool accuracy first rises then declines, indicating excessive entropy can confuse rewards. We selected $\alpha$=0.1, as it achieved the best balance between exploration and exploitation. Case study in Appendix A.5

**Generalizability of D-CORE.** We validate out-of-distribution generalization on ACEBench (Chen et al., 2025), $\tau^2$-Bench (Barres et al., 2025) and BFCLv4-agentic (Patil et al.). Table 4 shows that D-CORE remains competitive on new complex tasks, confirming that its gains stem from improved task decomposition reasoning and generalize well across benchmarks.

| Model | BFCLv3 MT(%) | $\tau$-bench Overall(%) |
|---|---|---|
| Qwen3-8B | 33.0 | 29.0 |
| +GRPO | 26.8(-6.2) | 31.5(+2.5) |
| +SD$_{n=10,000}$ | 51.5 | 22.1 |
| +SD$_{n=20,000}$ | 53.1 | 37.8 |
| +SD$_{n=40,000}$ | 57.5 | 36.6 |
| +GRPO | 67.4(+9.9) | 45.2(+8.6) |

Table 5: Effect of self-distillation on multi-turn tool use capability. SD: self-distillation; MT: Multi-Turn; n: number of training samples.

**Mitigating the Lazy Reasoning.** Figure 8b demonstrates that the D-CORE model reduces the proportion of Lazy Reasoning in incorrect answers for BFCLv3 Multi-Turn and $\tau$-bench to levels comparable with Parallel and Irrelevance categories. As a direct control, the 8B model shows a reduction in errors caused by Lazy Reasoning from 45% to 6% in Multi-Turn tasks, demonstrating the effectiveness of our approach in mitigating the Lazy Reasoning phenomenon.

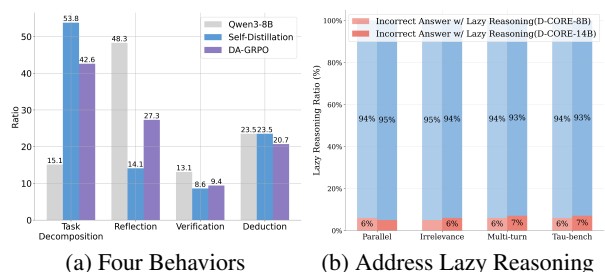

(a) Four Behaviors      (b) Address Lazy Reasoning

Figure 8: **(a)** Behavioral distribution changes after self-distillation and DA-GRPO. **(b)** Lazy Reasoning ratios across tasks for the D-CORE models.

**Effectiveness of Task Decomposition.** Table 7 shows the accuracy of Algorithm 1 on Qwen3-8B. Reference trajectories and few-shot examples improve task decomposition. However, ground truth trajectories in complex scenarios are often unavailable. Using pseudo-labels from Qwen3-Max for task decomposition, we demonstrate that our framework remains highly effective, confirming its practical applicability.

| Model | BFCLv3 overall(%) | $\tau$-bench overall(%) |
|---|---|---|
| DS-R1-Llama3.1-8B | 27.8 | 19.0 |
| D-CORE-Llama3.1-8B | 63.7 | 36.0 |
| DS-R1-Qwen2.5-14B | 47.9 | 18.2 |
| D-CORE-Qwen2.5-14B | 70.5 | 41.5 |

Table 6: Performance comparison of trajectory-distilled reasoning models: Deekseek-R1 vs. D-CORE (Qwen3-8B).

| Ref Traj | Few Shot | Traj Type | Success Rate(%) | $\tau$-bench overall(%) |
|---|---|---|---|---|
| ✗ | ✗ | ✗ | 73.8 | 20.3 |
| ✓ | ✗ | GT | 89.1 | 33.8 |
| ✓ | ✓ | GT | 93.2 | 37.1 |
| ✓ | ✓ | PL | 92.8 | 29.6 |

Table 7: Task decomposition success rates based on Qwen3-8B. GT: ground truth. PL: pseudo-labels.

**Quality of D-CORE Reasoning Trajectories.** To evaluate the effectiveness of D-CORE generated trajectories (Algorithm 1), we conduct SFT on Llama3.1-8B and Qwen2.5-14B using trajectories produced by Qwen3-8B. As shown in Table 6, models trained with our trajectories significantly outperform Deepseek-R1 SFT on complex tool-use scenarios.

## 5 RELATED WORK

**Tool Use.** Recent efforts have focused on creating and curating datasets to enhance LLMs tool use competencies (Patil et al., 2023; Liu et al., 2024a;b; Chen et al., 2024; Schick et al., 2023b). APIGen-MT (Prabhakar et al., 2025) and Magnet (Yin et al., 2025) all aim to enhance LLMs' multi-turn tool use capabilities by constructing datasets that closely resemble real-world scenarios. To tackle the poor generalizability of models trained using SFT, ToolRL (Qian et al., 2025) and Nemotron-N1 (Zhang et al., 2025) applied long CoT and RL training methodologies to tool use tasks and conducted evaluations on BFCLv3 (Patil et al.) single-turn tasks. However, the analysis and improvement of long CoT and RL's effectiveness in enhancing tool use within complex scenarios (Yao et al., 2024) remains a challenging problem.

**Large Reasoning Models.** Large Reasoning Models represent a significant advancement in the evolution of language models (OpenAI, 2024b; 2025; Anthropic, 2025; Yang et al., 2025). DeepSeek-R1 (Guo et al., 2025) demonstrates that the GRPO (Shao et al., 2024) optimization algorithm combined with outcome reward mechanisms can enhance models' reasoning capabilities. A series of works have analyzed the impact of the reasoning process on outcomes from various perspectives (Ning et al., 2025; Hu et al., 2025; Gandhi et al., 2025; Shojaee et al., 2025). Additionally, some research on task-specific reasoning approach (Khot et al., 2022a; Zhou et al., 2022; Yao et al., 2023a;b; Besta et al., 2024) and Tool-Integrated Reasoning (Li et al., 2025a; Dong et al., 2025; Li et al., 2025b)has regained attention for improving current LRM training. Our work systematically transfers advances in reasoning to complex tool use for LRM, leading to strong empirical gains.

## 6 CONCLUSION

In this paper, We systematically study LRMs on tool use tasks. While reasoning boosts tool use, it often devolves into "Lazy Reasoning" phenomenon in complex tool use scenarios. We conducted a detailed analysis and identified that the root cause of this phenomenon is a compensatory mechanism arising from the LRMs' insufficient task decomposition capabilities. To address this, we proposed the D-CORE method, which uses self-distillation and DA-GRPO to consistently enhance LRMs' performance on complex tool use tasks. LRMs trained with D-CORE achieve state-of-the-art performance on BFCL-v3 and $\tau$-bench tasks. Our promising results demonstrate considerable potential for generalization and scalability, achieved within the limits of current resources. Future work will focus on applying our method to Mixture of Experts (MoE) architectures and multimodal reasoning models. Additionally, we plan to build on our current research by introducing more complex reinforcement learning algorithms to further explore more efficient reasoning for tool use.

## ETHICS STATEMENT

This work adheres to the ICLR Code of Ethics. In this study, no human subjects or animal experimentation was involved. All datasets used, including trajectories of our custom-build tool use agent, were sourced in compliance with relevant usage guidelines, ensuring no violation of privacy. We have taken care to avoid any biases or discriminatory outcomes in our research process.

## REPRODUCIBILITY STATEMENT

We have made every effort to ensure that the results presented in this paper are reproducible. All code and models will be open sourced to facilitate replication and verification. The key experiments mentioned in the motivation section have been included in the supplementary materials to demonstrate that our findings are reproducible.

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

# A  APPENDIX

## A.1  DETAILS OF LAZY REASONING

**Case Study of Lazy Reasoning.**    The following cases clearly demonstrate the Lazy Reasoning phenomenon exhibited by Qwen3-8B and Qwen3-32B when completing tasks on $\tau$-bench. This manifests as the models' lack of detailed planning for the subtasks required to complete the main task, as well as the absence of specific reasoning for each subtask execution. Instead, the models display an inefficient, random pattern of trial-negation-retry thinking. Such reasoning processes occur in both correct and incorrect responses, indicating a cognitive pattern that models adopt when solving tool-use tasks, rather than an accidentally triggered behavior.

---

**Lazy Reasoning in Qwen3-8B**

```
<think>
```
Okay, let's tackle the user's question. They want two things: the precise mileage from San Francisco to Rivermist and an assessment of whether their current fuel will suffice for the journey.

First, I need to check if there's a function to estimate the distance between two cities. Looking at the tools provided, there's 'estimate distance' which takes cityA and cityB as parameters. But wait, the description says it uses zipcodes. The user provided city names, not zipcodes. So I might need to get the zipcodes first. There's a function called 'get zipcode based oncity' that can get a zipcode from a city name. So first, I should call that for both San Francisco and Rivermist. Once I have the zipcodes, I can use 'estimate distance' to get the distance. Then, the user needs to know if their current fuel is enough. The 'estimate drive feasibility by mileage' function takes a distance in m iles and checks if the vehicle can drive that distance. But I need the distance in miles. The 'estimate distance' function returns the distance in km, so I might need to convert that to miles. Wait, there's a 'gallon to liter' and 'liter to gallon' function, but not km to miles. Hmm, the user's question is about fuel sufficiency, which probably depends on the vehicle's fuel efficiency. Wait, the 'estimate drive feasibility by mileage' function's description says it estimates mileage based on distance. Maybe it uses the vehicle's fuel efficiency internally. But the user's current fuel level isn't provided. The 'get current speed' function exists but not fuel level. Wait, there's 'displayCarStatus' which includes fuel Level. So maybe I need to check the current fuel level first. But the user hasn't provided that info. Wait, the user is asking if their current fuel should suffice, so the function might need the distance and the current fuel level. But the 'estimate drive feasibility by mileage' function only takes distance. Hmm, maybe the function assumes the vehicle's fuel efficiency is known, and checks if the distance is within the range based on current fuel. But without knowing the fuel efficiency, how can it calculate? Maybe the function has internal data. Alternatively, maybe the user's current fuel level is required as a parameter, but it's not in the function's parameters. Wait, looking at the functions again: 'estimate drive feasibility by mileage' requires 'distance' in miles. The response is whether the vehicle can drive that distance. But without knowing the fuel level, how can it determine that? Maybe there's another function to get the fuel level. There's 'display CarStatus' which includes fuel Level. So perhaps the process is: get the distance, convert km to miles, then check feasibility using the distance and the current fuel level. But the function 'estimate drive feasibility by mileage' doesn't take fuel level. This seems like a gap. Wait, maybe the function uses the vehicle's fuel efficiency internally, so if the user provides the distance, it calculates based on the vehicle's known efficiency. But since the user's fuel level isn't provided, maybe the function can't check if the fuel is sufficient. This is a problem. The user's question requires both the distance and fuel sufficiency assessment, but the a vailable functions might not have all the necessary parameters. However, the user might have to proceed with the available functions. Let's proceed step by step.
```
</think>
```

---

**Lazy Reasoning in Qwen3-32B**

```
<think>
```
Okay, the user wants to book a flight from San Francisco to Los Angeles. They mentioned the travel class is 'first' and the date is '2024-11-15'. They also provided the card ID 'card123' and access token 'access token abc123'. First, I need to get the nearest airports for both cities. The previous steps already did that using get nearest airport by city, which returned SFO and LAX. So the travel from is SFO and travel to is LAX. Next, the book flight function requires several parameters: access token, card id, travel date, travel from, travel to, travel class, and travel cost. The user provided access token abc123 and card123. The date, from, to, and class are given. But the travel cost isn't mentioned. Wait, the user didn't specify the cost. However, the get flight cost function can retrieve the cost based on the parameters. But the user might not have provided that info. Wait, the book flight function requires all parameters, including travel cost. Since the user didn't provide it, but the assistant might need to infer or fetch it. However, in the given functions, there's get flight cost which returns a list of costs. But the user might have intended to use that. However, the user hasn't mentioned checking the cost, just to book it. So maybe the assistant should proceed with the booking, but the travel cost is a required parameter. Since the user didn't provide it, perhaps the assistant should point that out. But the user's message says "nag that booking for me," implying that the cost is already known or acceptable. Wait, maybe the get flight cost is not necessary here because the user is directly booking. But the book flight function requires the travel cost. Since the user hasn't provided it, the assistant can't proceed. However, in the initial message, the user might have missed including the cost. Alternatively, maybe the get flight cost is called implicitly. But according to the functions provided, the book flight requires all parameters, including travel cost. Therefore, the assistant needs to get the travel cost first. But the user hasn't provided it. So the assistant should call get flight cost with the given parameters to retrieve the cost, then use that to book. However, the user's instruction is to book the flight, so maybe the assistant should proceed with the booking, but since the cost is required, it's a problem. Alternatively, maybe the assistant can proceed without the cost, but the function requires it. Therefore, the correct approach is to first get the flight cost using get flight cost, then use that cost in the book flight function. But the user hasn't mentioned checking the cost, so maybe the assistant should proceed with the booking, but the function requires the cost. Since the user hasn't provided it, the assistant can't proceed. However, the user might have intended to include the cost in their message but forgot. Alternatively, maybe the cost is not required, but according to the function definition, it is. Therefore, the assistant must call get flight cost first to obtain the travel cost. Let's proceed with that. So first, call get flight cost with travel from=SFO, travel to=LAX, travel date=2024-11-15, travel class=first. Then, once the cost is retrieved, use it in book flight. But the user's message doesn't mention checking the cost, so maybe the assistant should proceed to book with the given parameters, but since the cost is missing, it's an error. However, the user might have intended to provide all necessary parameters. Wait, looking back, the user's message says I've got my card with id 'card123' and account token 'access token abc123' all good to go.They provided all parameters except travel cost. Therefore, the assistant needs to get the travel cost first. So the next step is to call get flight cost with the given parameters. Then, once the cost is known, proceed to book. Therefore, the correct function call is get flight cost with the parameters travel from=SFO, travel to=LAX, travel date=2024-11-15, travel class=first. So the next function call should be get flight.
```
</think>
```

### A.1.1 ANALYSIS OF LAZY REASONING.

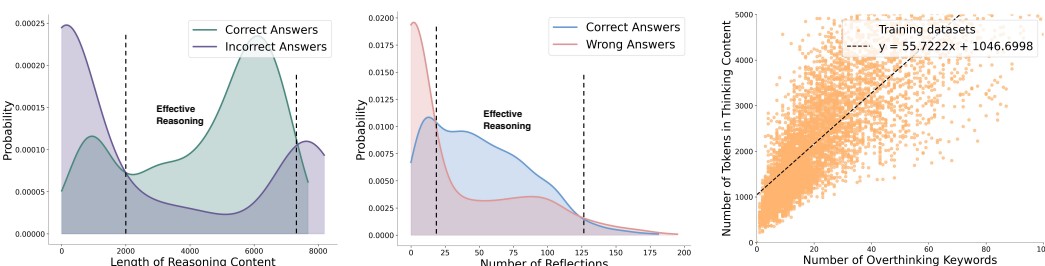

Figure 9: Distribution analysis of reasoning processes from Qwen3-8B rollout experiments on the MATH dataset. (a) Probability density functions of reasoning process lengths for correct vs. incorrect answers. (b) Probability density functions of reflection counts in reasoning processes for correct vs. incorrect answers. (c) Distribution and fitted function of reflection words versus reasoning length.

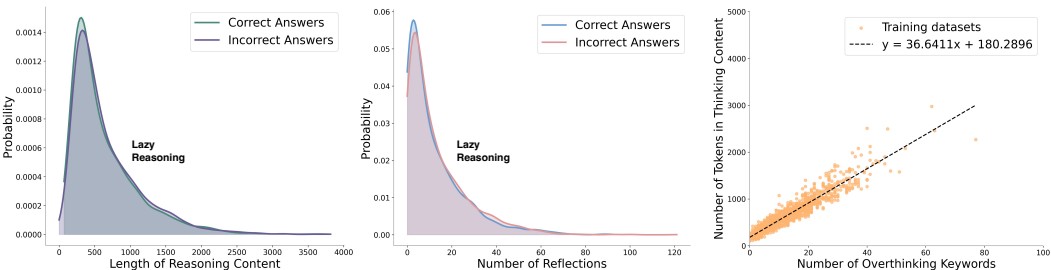

Figure 10: Distribution analysis of reasoning processes from Qwen3-8B rollout experiments on the BFCLv3 multi-turn dataset. (a) Probability density functions of reasoning process lengths for correct vs. incorrect answers. (b) Probability density functions of reflection counts in reasoning processes for correct vs. incorrect answers. (c) Distribution and fitted function of reflection words versus reasoning length.

Figure 9 and Figure 10 present statistical analyses of rollout sampling experiments conducted with Qwen3-8B across MATH and BFCLv3 multi-turn tasks, using cases with exactly 50% accuracy (10 correct, 10 incorrect of 20 rollouts). For MATH tasks, increased reasoning length and reflection frequency create an effective reasoning region where correct responses dominate, proving reasoning's value. However, BFCLv3 multi-turn tasks show highly consistent distributions between correct and incorrect answers, with no effective reasoning region. This phenomenon suggests two plausible explanations: (1) the complex tool use task requires no reasoning, or (2) the model's reasoning provides no benefit for complex tool use, indicating fundamental issues with the current reasoning approach. While the first explanation presents an intriguing possibility, we maintain that reasoning should enhance tool use accuracy based on our single-turn experimental evidence—otherwise, this would create a fundamental contradiction with established findings.

We hypothesize substantial improvement potential in Qwen3-8B's complex tool use reasoning. Between RL-based refinement and SFT-based knowledge injection, we pursue SFT after RL proved ineffective—unsurprisingly, given that current RL primarily reinforces existing patterns. This raises a fundamental question: what constitutes effective tool use reasoning? To narrow our research scope, we propose a key hypothesis—LRMs may inherently possess tool use capabilities that are somehow constrained, supported by their strong single-turn performance.

To validate this conjecture, we initially attempted direct intervention by inserting desired reasoning patterns within `<think>` and `</think>`, but this failed catastrophically, reducing tool-calling accuracy to near-zero. Since LRM reasoning is shaped by RL optimization, instruction-following within reasoning blocks remains unoptimized. Given this constraint, query modification becomes the only viable intervention point, naturally motivating the D-CORE framework.

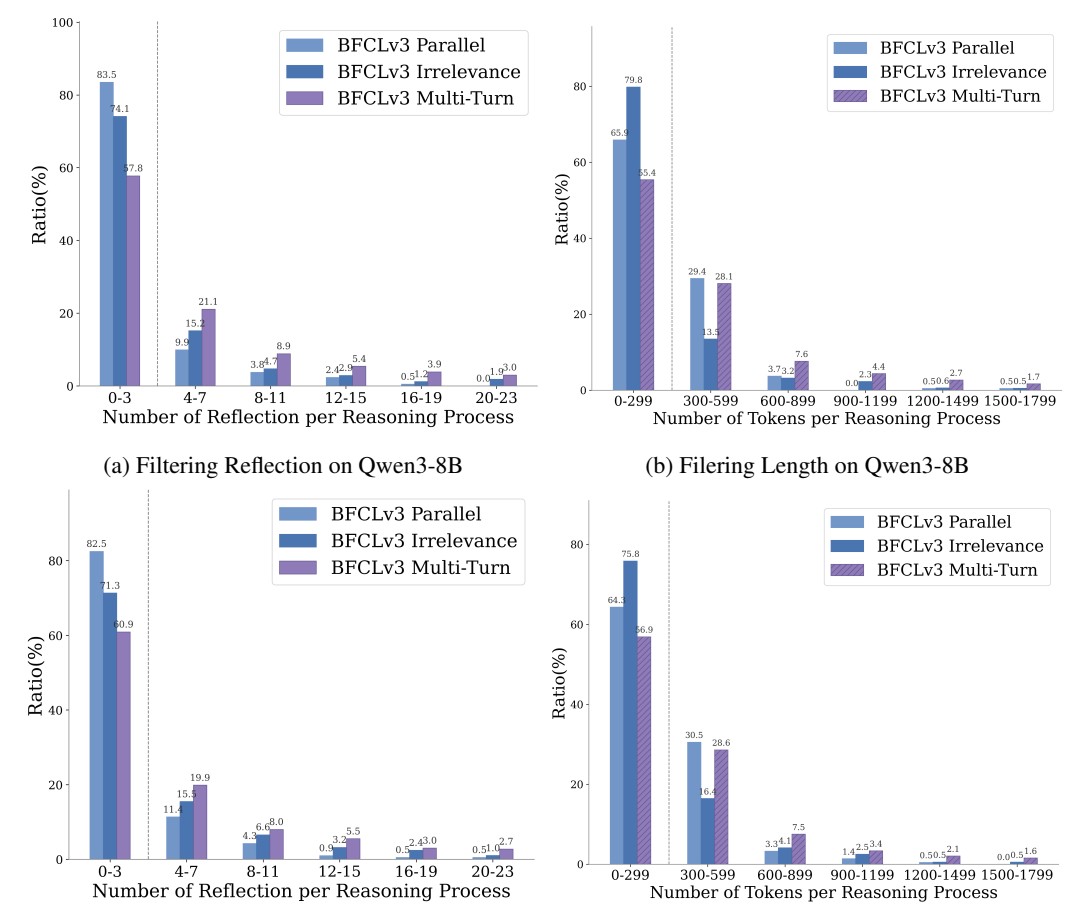

(a) Filtering Reflection on Qwen3-8B      (b) Filering Length on Qwen3-8B

(c) Filtering Reflection on Qwen3-32B      (d) Filering Length on Qwen3-32B

Figure 11: **(a)** Filtering Lazy Reasoning from the reasoning processes of Qwen3-8B on BFCLv3 using reflection threshold-based filtering. **(b)** Filtering Lazy Reasoning from the reasoning processes of Qwen3-8B on BFCLv3 using token number threshold-based filtering **(c)** Filtering Lazy Reasoning from the reasoning processes of Qwen3-32B on BFCLv3 using reflection threshold-based filtering. **(d)** Filtering Lazy Reasoning from the reasoning processes of Qwen3-32B on BFCLv3 using token number threshold-based filtering.

### A.1.2 FILTERING LAZY REASONING.

After observing the Lazy Reasoning phenomenon, we first analyze the reasoning processes of Qwen3-8B and Qwen3-32B models on BFCLv3. We use reflection keywords to count the number of reflections contained in each reasoning process, and then compile histograms as shown in Figure 11. We define reasoning processes with more than 300 tokens and containing over 3 reflections as demonstrating Lazy Reasoning behavior, and apply this common filtering threshold across all tasks to ensure fairness. As can be observed, within each histogram interval, the Qwen3 models generate more reflections in Multi-Turn scenarios. Finally, we calculate the proportion of incorrect answers containing Lazy Reasoning relative to all answers, which yields Figure 3c presented in the main text.

### A.1.3 WHY LRMS HAVE LAZY REASONING?

The question of why models exhibit Lazy Reasoning is both straightforward and complex. It is straightforward because models develop this phenomenon when encountering specific problems due to the influence of optimization methods and data distribution during the Supervised Fine-Tuning (SFT) or Reinforcement Learning (RL) stages. It is complex because current open-source models do not publicly disclose their training datasets and optimization approaches, and even when such information is available, few organizations possess sufficient resources to conduct comprehensive debugging research to determine at which stage and through what type of data introduction this

phenomenon emerges. However, we reveal one possible cause of this phenomenon: models lack critical thinking patterns for certain specific problems. This finding advocates that future reasoning model training should consider adopting different thinking approaches for different tasks.

### A.1.4 Do all LRMs have Lazy Reasoning?

The answer is yes. As defined in the main text, Lazy Reasoning is essentially a posterior concept. Whenever a LRM demonstrates low accuracy on a specific problem or category of problems, and engages in extensive ineffective reflection in the incorrect cases, the model exhibits Lazy Reasoning on those problems. This phenomenon can certainly be detected through testing, as current LRMs cannot demonstrate clear and effective reasoning processes across all problems in the world—while this remains a shared aspiration among algorithm researchers, it would require enormous costs to achieve. As demonstrated in our filtering experiments, we can always identify such patterns using appropriate thresholds and certain metrics. More importantly, this phenomenon reveals that when training reasoning models, we need to inject some a prior problem-solving approaches for specific problems into the LRM. This approach can significantly help LRM avoid optimization difficulties during the RL process.

### A.1.5 Case Study of Addressing Lazy Reasoning

We employ prompts to guide the model in decomposing queries into subtasks. We then manually replace all queries in the tasks with these decomposed subtasks and sequentially collect the execution results of each subtask as the model's response to the original query. This approach significantly mitigates Lazy Reasoning. Figure 12 demonstrates the workflow of using Qwen3-8B with Decomposed Prompting on BFCLv3 Multi-Turn Base task. As shown, the Qwen3-8B model requires 1,616 tokens for reasoning on original query. However, after decomposition into three subtasks, the reasoning lengths for each subtask are 367, 185, and 247 tokens respectively, totaling 799 tokens—only half of the original reasoning length. Moreover, the aggregated responses from the three subtasks form the correct answer. This case study provides compelling evidence for the feasibility of

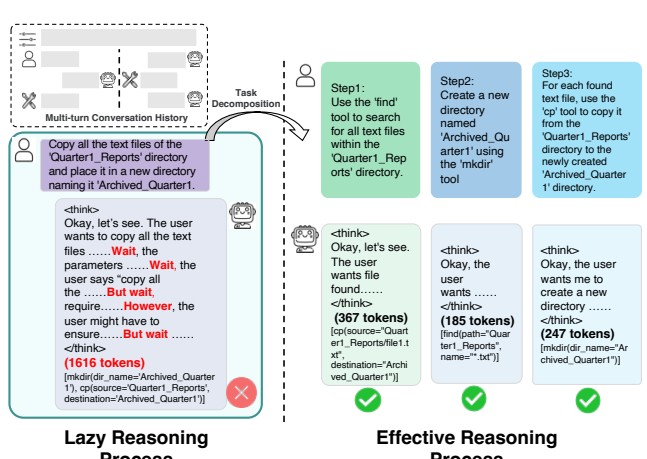

Figure 12: Workflow of converting Lazy Reasoning process to Effective Reasoning process using decomposed prompting. The model is Qwen3-8B, and the example is from the first question of BFCLv3 Multi-Turn Base task.

using Decomposed Prompting to alleviate Lazy Reasoning. Naturally, this decomposition approach does not achieve a 100% success rate. We address this limitation in our D-CORE algorithm by refining the decomposition mechanism through the addition of ground truth and few-shot examples, thereby substantially improving the approach's feasibility.

## A.2 TASK DECOMPOSITION PROMPTS

---

**Task Decomposition Prompt**

```
You are a task decomposition expert. Now you need to reverse-
engineer the process of breaking down complex queries into subtasks
based on the given information.

###Input Information:
1. System Policy:[Insert the system policy here]
2. Available Tool List:[{"name": Tool Name 1, "description":
    Function description}, {"name": Tool Name 2, "description":
    Function description}...]
3. Chat History:[Insert the chat history here]
4. Query:[Insert the query here]
5. Final Tool Invocation Results:[Call Tool A, Call Tool B]
...

## Task Requirements:
Based on the above information, please reverse-engineer a
reasonable subtask decomposition process based on Query and Chat
History. Just output the subtask list in following format.Do not
include information in the subtask description that does not exist
in the chat history and query.

## Output Format:
[{"step":1 , "description": Subtask 1}, {"step": 2, "description":
    Subtask 2}...]

<example_1>
...
</example_1>

<example_2>
...
</example_2>

<example_3>
...
</example_3>

Please begin the analysis:
```

---

## A.3 REASONING GENERATION AND COMPOSITION

### A.3.1 SEQUENTIAL REASONING GENERATION AND COMPOSITION.

**Sequential Reasoning Composition Template**

```python
def reasoning_processes_composition_sequential(task, subtasks,
    reasoning_processes):
    reasoning_list = []
    answer_list = []
    full_reasoning = f"OK, let me analyze the overall task: {task}\
n"
    # Add reasoning process for each subtask
    for i, (subtask, reasoning_process) in enumerate(zip(subtasks,
    reasoning_processes), 1):
        if i == 1:
            full_reasoning_step = full_reasoning + f"Now I should
    analyze the execution process of subtask {i}: {subtask}\n"
        else:
            full_reasoning_step = full_reasoning + f"I have already
     solve the subtask:\n"
            for j in range(1, i):
                full_reasoning_step += f"{j}.{subtasks[j-1]}\n"
            full_reasoning_step += f"Now I should analyze the
    execution process of subtask {i}:{subtask}\n"
        # Extract reasoning thinking process (remove tags and
    function call parts)
        thinking_part = reasoning_process[0].replace("", "").
    replace("", "").strip()
        thinking_part = thinking_part.split('<think>\n')[-1]
        full_reasoning_step += f"{thinking_part}\n"
        reasoning_list.append(full_reasoning_step)
        answer_list.append(reasoning_process[1])
    return reasoning_list, answer_list
```

### A.3.2 PARALLEL REASONING GENERATION AND COMPOSITION.

**Parallel Reasoning Composition Template**

```python
def reasoning_processes_composition_parallel(task, subtasks,
    reasoning_processes):
    full_reasoning = f"OK, let me analyze the overall task: {task}\
    n To complete this task, I need to break it down into the
    following subtasks:\n"
     # Add subtask list
    for i, subtask in enumerate(subtasks, 1):
        full_reasoning += f"{i}. {subtask}\n"

    full_reasoning += "Wait, I discovered that there are no
    dependencies between these subtasks, meaning that no subtask's
    input requires the output from another subtask's execution, so I
     can run these subtasks in parallel.\n"

    # Add reasoning process for each subtask
    for i, (subtask, reasoning_process) in enumerate(zip(subtasks,
    reasoning_processes), 1):
        full_reasoning +=  f"Now I should analyze the execution
    process of subtask {i}: {subtask}\n"

        thinking_part = reasoning_process[0].replace("", "").
    replace("", "").strip()
        thinking_part = thinking_part.split('<think>\n')[-1]
        full_reasoning += f"{thinking_part}\n"

    answer_list = []
    for step in reasoning_processes:
        answer_list.append(step[1].strip())
    composed_answer = combine_bracketed_strings(answer_list)
    full_reasoning = full_reasoning + "Finally, I can output a
    parallel tool call result, and after verification, this result
    meets the user's requirements:" + composed_answer
    return full_reasoning, composed_answer
```

### A.3.3 IRRELEVANCE REASONING GENERATION AND COMPOSITION.

**Irrelevance Reasoning Composition Template**

```python
def irrelevance_reasoning_generation_and_composition_(tasks):
    trajectories = []
    for task in tqdm.tqdm(tasks):
        conversations = copy.deepcopy(task['conversations'])
        real_answer = conversations[-1]['content'].split('</think>\
n\n')[-1]
        conversations.pop()
        reasoning, answer = request_reasoning_model(conversations)
        full_reasoning = f"OK, let me analyze the overall task: {
    task}\n However, I find that the user's query is missing key
    information, or the user's query cannot be fulfilled with the
    current list of functions. I need to conduct a detailed analysis
     of this.\n"

        full_reasoning += reasoning
        irr_query = {'role':'assistant', 'content':f"<think>\n\n{
    full_reasoning}</think>\n\n{real_answer}"}
        irr_conversations = copy.deepcopy(task['conversations'])
        irr_conversations.pop()
        irr_conversations.append(finish_query)
        trajectories.append({'conversations': irr_conversations, '
    id': i})
    return trajectories
```

A.4 ENTROPY BASED ADVANTAGE.

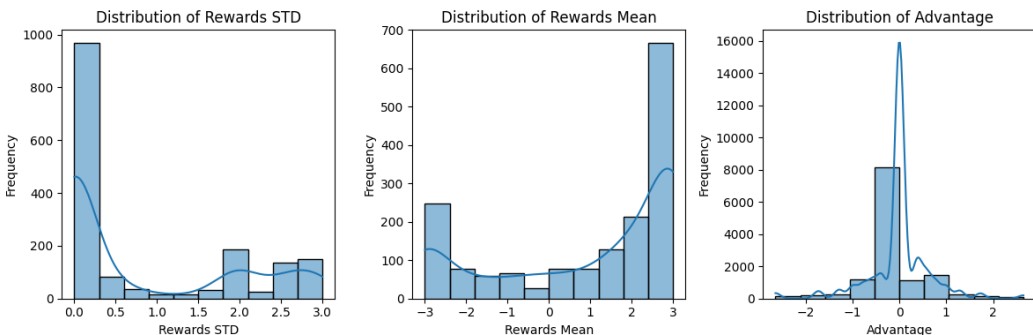

Figure 13: Distribution of Standard Deviation, Mean, and Advantage during GRPO Training on Qwen3-8B at step 0.

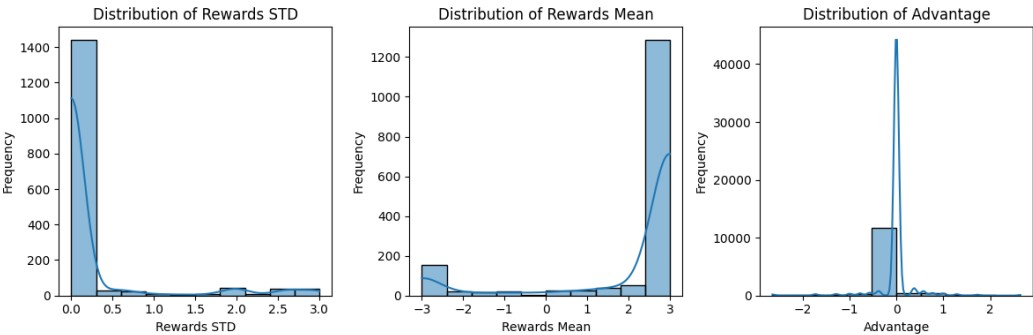

Figure 14: Distribution of Standard Deviation, Mean, and Advantage during GRPO Training on Self Distillation Qwen3-8B at step 0.

Figures 13 and 14 present the reward distributions at step 0 for the complex tool use dataset when performing GRPO with the original Qwen3-8B model and the self-distilled Qwen3-8B model, respectively. The results demonstrate that the self-distilled model exhibits more concentrated mean and variance in rewards, leading to advantage distributions heavily clustered around zero. This phenomenon indicates that self-distillation has already enhanced the model's capability for tool use, resulting in rollouts with high accuracy and consistency on RL training datasets. Consequently, numerous groups emerge with a mean reward of 3 and 0 variance. When these groups are processed through the advantage function, their collective advantage become zero.

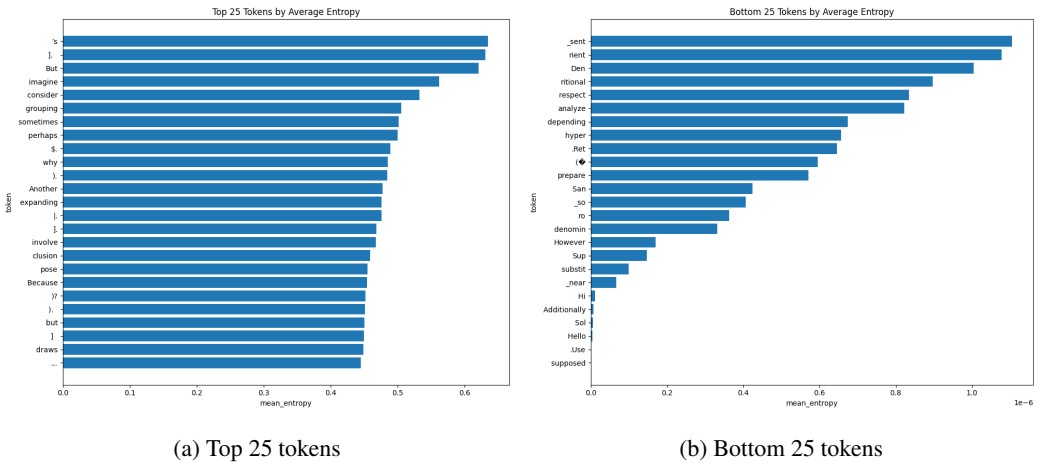

(a) Top 25 tokens           (b) Bottom 25 tokens

Figure 15: **(a)** Top 25 tokens with highest average entropy. **(b)**Bottom 25 tokens with lowest average entropy.

Figure 15 displays the 25 tokens with the highest average entropy, and Figure 10b shows the 25 tokens with the lowest average entropy. From the distribution of these tokens, we can observe that high-entropy tokens contain words like `'but,'` `'perhaps,'` and `'because.'`. Based on these observations, combined with recent research developments (Cheng et al., 2025), we propose incorporating the average entropy of each rollout into the advantage function. This modification prevents advantages from clustering around zero and guides the model toward exploring tokens with higher entropy.

### A.5 GRPO FAILURE CASE STUDY

Below are three case studies comparing D-CORE-8B trained with GRPO, DA-GRPO ($\alpha$=0.1, $\sigma$=0.5), and DA-GRPO ($\alpha$=1.0, $\sigma$=1.0). The case is the 28th sample from the retail task in $\tau$-bench, where the user requests to return order `W7181492`. This order contains two items—`"5753502325"` and `"9851293632"`—thus requiring the model to invoke the `return delivered order items` function with parameter `item ids=["5753502325", "9851293632"]`. GRPO answered incorrectly with `item ids="5753502325"`, omitting `"9851293632"`. The underlying reason is that the model decomposed the task into 6 subtasks. In contrast, DA-GRPO with $\alpha$=0.1 successfully answered this question by rewarding high-entropy tokens, enabling the model to learn diverse reasoning processes and break free from the default 6-subtask reasoning path, leading to correct `item ids=["5753502325", "9851293632"]`. However, under the setting of $\alpha$=1.0 and $\sigma$=1.0, DA-GRPO introduces excessive high-entropy token advantages that interfere with the original reward advantage, biasing the model's reasoning process toward generating more high-entropy tokens such as `"wait"`, `"but"`, and `"however"`, ultimately leading to incorrect answers. This comparison clearly demonstrates how DA-GRPO diversifies model reasoning by rewarding high-entropy tokens, and how excessive reflection leads to failure when high-entropy token advantages dominate.

---

**GRPO**

***Ground truth***:

```
{
  "name": "return_delivered_order_items",
  "arguments": "{"order_id": "#W7181492",
  "item_ids": ["5753502325", "9851293632"],
  "payment_method_id": "paypal_3024827"}"
}
```

***Reasoning Process***:

OK, let me analyze the overall task: Yes, and the reason is ordered by mistake.

To complete this task, I need to break it down into the following subtasks:

1. Process the return for the skateboard (Item ID: `4293355847`) from order `#W3792453` using the original payment method (PayPal).

2. Process the return for the garden hose (Item ID: `5753502325`) from order `#W7181492` using the original payment method (PayPal).

3. Process the return for the backpack (Item ID: `9851293632`) from order `#W7181492` using the original payment method (PayPal).

4. Process the return for the mechanical keyboard (Item ID: `9570044148`) from order `# W5565470` using the original payment method (PayPal).

5. Process the return for the pet bed (Item ID: `6857426243`) from order `#W5565470` using the original payment method (PayPal).

6. Cancel the garden hose order (Order ID: `#W2575533`) with the reason 'ordered by mistake'.

 I have already solve the subtask:

1.Process the return for the skateboard (Item ID: `4293355847`) from order `#W3792453` using the original payment method (PayPal).

Now I should analyze the execution process of subtask 2:

Process the return for the garden hose (Item ID: `5753502325`) from order `#W7181492` using the original payment method (PayPal).

Okay, let's see. The user wants to return the garden hose from order `#W7181492`. First, I

---

need to check if that order is delivered. From the previous tool responses, when we checked order `#W7181492`, the status was "delivered". So, returning items from a delivered order is allowed.

The function to use here is `return_delivered_order_items`. The parameters required are `order_id`, `item_ids`, and `payment_method_id`. The user provided the order ID as `#W7181492`, the item ID as 5753502325, and the payment method is PayPal, which has the ID `paypal_3024827`.

I should make sure that the item ID exists in the order. Looking back at the order details for `#W7181492`, yes, the garden hose has item ID `5753502325`. The payment method is the original one, which is correct as per the user's instruction.

No other parameters are needed, and the function call should be straightforward. Just need to format it correctly with the right parameters.

---

### DA-GRPO with $\alpha=0.1$, $\sigma=0.5$

***Ground truth***:

```
{
  "name": "return_delivered_order_items",
  "arguments": "{"order_id": "#W7181492",
  "item_ids": ["5753502325", "9851293632"],
  "payment_method_id": "paypal_3024827"}"
}
```

***Reasoning Process***:

OK, let me analyze the overall task: Okay. Just process the returns and tell me the total refund amount.

To complete this task, I need to break it down into the following subtasks:

1. Process the return for order `#W3792453` by calling `return_delivered_order_items` with the item ID of the skateboard and using the original payment method (`paypal_3024827`) for the refund.

2. Process the return for order `#W7181492` by calling `return_delivered_order_items` with the item IDs of the **garden hose and backpack**, and using the original payment method (`paypal_3024827`) for the refund.

3. Process the return for order `#W5565470` by calling `return_delivered_order_items` with the item IDs of the mechanical keyboard and pet bed, and using the original payment method (`paypal_3024827`) for the refund.

I have already solve the subtask:

1.Process the return for order `#W3792453` by calling `return_delivered_order_items` with the item ID of the skateboard and using the original payment method (`paypal_3024827`) for the refund.

Now I should analyze the execution process of subtask 2:

Process the return for order `#W7181492` by calling `return_delivered_order_items` with the item IDs of the garden hose and backpack, and using the original payment method (`paypal_3024827`) for the refund.

Okay, let's see. The user wants to return items from order `#W7181492`. **The items are the garden hose and backpack**. I need to check the order details again to make sure those item IDs are correct.

Looking back, order `#W7181492` has items with IDs `5753502325` (Garden Hose) and `9851293632` (Backpack). The user specified those item IDs, so that's correct. The payment method is PayPal, which is the original method.

I should call the `return_delivered_order_items` function with `order_id=#W7181492`, `item_ids=[5753502325, 9851293632]`, and `payment_method_id=paypal_3024827`.

Need to make sure the parameters are in the right format and required fields are included. All required parameters are present, so the function call should work.

---

### DA-GRPO with $\alpha=1.0$, $\sigma=1.0$

*Ground truth*:

```
{
  "name": "return_delivered_order_items",
  "arguments": "{"order_id": "#W7181492",
  "item_ids": ["5753502325", "9851293632"],
  "payment_method_id": "paypal_3024827"}"
}
```

*Reasoning Process*:

OK, let me analyze the overall task: Yes.

To complete this task, I need to break it down into the following subtasks:

1. Process the return for the specified items from order `#W3792453`, `#W7181492`, and `#W5565470` by calling `return_delivered_order_items`. The items to return are: skateboard (item ID: `4293355847`), garden hose from order `#W7181492` (item ID: `5753502325`), backpack from order `#W7181492` (item ID: `9851293632`), mechanical keyboard from order `#W5565470` (item ID: `9570044148`), and pet bed from order `#W5565470` (item ID: `6857426243`). The refund will be processed to the user's PayPal account (`payment_method_id: paypal_3024827`).

2. Cancel the pending order `#W2575533` by calling `cancel_pending_order`, as the user no longer needs the garden hose (item ID: `5206946487`) and the order is in 'pending' status.

Now I should analyze the execution process of subtask 1:

Process the return for the specified items from order `#W3792453`, `#W7181492`, and `#W5565470` by calling `return_delivered_order_items`. The items to return are: skateboard (item ID: `4293355847`), garden hose from order `#W7181492` (item ID: `5753502325`), backpack from order `#W7181492` (item ID: `9851293632`), mechanical keyboard from order `#W5565470` (item ID: `9570044148`), and pet bed from order `#W5565470` (item ID: `6857426243`). The refund will be processed to the user's PayPal account (`payment_method_id: paypal_3024827`).

Okay, let's see. The user wants to return several items from different orders. The orders mentioned are `#W3792453`, `#W7181492`, and `#W5565470`. The items to return are the skateboard from `#W3792453`, the garden hose and backpack from `#W7181492`, the mechanical keyboard and pet bed from `#W5565470`. The refund should go to their PayPal account.

First, I need to check if these orders are in the 'delivered' status because returns can only be processed for delivered orders. From the previous interactions, the status of `#W3792453` was delivered, `#W7181492` was delivered, and `#W5565470` was also delivered. So that's good.

Next, the function required for returning items is `return_delivered_order_items`. The parameters needed are `order_id`, `item_ids`, and `payment_method_id`. However, the function seems to require a single `order_id` per call. The user has items from three different orders.

Wait, looking at the function definition again: the `return_delivered_order_items` function takes an `order_id`, which suggests that each call is for a single order. So, if the items are from different orders, I need to make multiple calls. But the user's instruction says to call `return_delivered_order_items` once. That might not be possible because the function expects a single order ID each time.

But maybe the user expects a single call, but according to the function's parameters, each call is per order. So, perhaps I need to make three separate calls: one for each order. However, the user's message says \"call `return_delivered_order_items`\" once, which might be a mistake. Alternatively, maybe the function allows multiple item IDs across different orders in one call, but the parameters don't specify that. The function's parameters have `order_id` as a string, `item_ids` as an array, but the `order_id` is per order.

So, the correct approach would be to split the items into their respective orders and make separate function calls. For example:

1. For order `#W3792453`, return the skateboard (item ID `4293355847`).
2. For order `#W7181492`, return the garden hose (`5753502325`) and backpack (`9851293632`).
3. For order `#W5565470`, return the mechanical keyboard (`9570044148`) and pet bed

> (6857426243).
>
> Each of these would require a separate call to `return_delivered_order_items` with the respective `order_id` and `item_ids`. The `payment_method_id` would be the same (`paypal_3024827`) for all, as specified.
>
> However, the user's instruction says to process the return by calling the function once. This might be an error, but given the function's parameters, it's not possible to handle all items in a single call. Therefore, I should proceed by making multiple function calls as needed.

## A.6 EXPERIMENTAL DETAILS

### A.6.1 SELF-DISTILLATION

In the data generation phase described in Algoritm 1, we deploy Qwen3-8B and Qwen3-14B models using vLLM 0.8.2 on a single 80G A100 GPU. For the Qwen3-8B model, it takes 25 hours to generate 40,000 data samples, while the Qwen3-14B model requires 30 hours to generate 40,000 training samples.

At the SFT stage, all of our experiments are performed on 8×80G A100 GPUs, using the Qwen3 model as the base for each independent experiment. We utilized the Huggingface Transformers library with version 4.51.3 to execute our training. During training, we employed DeepSpeed Zero3 optimization and Flashattention-2 to enhance memory efficiency. Learning rate is 1.0e-05, using cosine learning rate scheduler type, max context size during training is 16384. Batch size is 2 per GPU. Specifically, when processing multi-turn training data, we split each multi-turn tool-use sample into multiple samples based on the number of turns. For these split samples, we only compute the loss for the final turn's answer in each sample, excluding the loss calculation for intermediate turns' answers. We then pack multiple samples into a single sample of length 16,384 using packing techniques to accelerate training. Additionally, the packed samples employ attention mask isolation to prevent contamination. The training time is 11 hours for Qwen3-8B and 21 hours for Qwen3-14B model, with both models trained for 3 epochs.

### A.6.2 DIVERSITY-AWARE REINFORCEMENT LEARNING

Our reinforcement learning training uses verl version 0.5.0, with all experiments conducted on 8×80G A100 GPUs. We set the advantage clipping ratios (adv clip ratio low, adv clip ratio high, and adv clip ratio) to 0.2. For KL divergence settings, we disable KL usage in reward calculation (use kl in reward=False) and set kl coef to 0.0, as GRPO employs KL loss in the actor rather than the reward. We enable KL loss in the actor (use kl loss=True) with a coefficient of 0.001 and use the 'low var kl' loss type. The maximum prompt length is set to 8,192 tokens and maximum response length to 4,096 tokens, with overlong prompts filtered out. For actor rollout sampling, we use a temperature of 1.0, top p of 1.0, top k of -1 (indicating vLLM rollout), and validation top p of 0.7. For diversity-aware settings, both Qwen3-8B and Qwen3-14B models are trained with $\alpha = 0.1$ and $\sigma = 0.5$. The training time is 11 hours for Qwen3-8B and 17 hours for Qwen3-14B model, with both models trained for 3 epochs.

## A.7 QUALITATIVE COMPARISON OF REASONING TRAJECTORIES

### A.7.1 CASE STUDY ON BFCLV3-MULTI-TURN 196

---

**Qwen3-8B**

{**"role"**: "user", **"content"**: "Gearing up for some exciting travel adventures! I wish to put a firm budget in place, setting a cap equivalent to 50,000 RMB for my upcoming European excursions. I'd appreciate if you could take charge of the conversion and establish a budget framework for me using the access token 'abc123xyz'."}

{**"role"**: "assistant", **"content"**: "`<think>` Okay, let's see. The user wants to set a budget limit of 50,000 RMB for their European travels and needs the conversion handled using the access token 'abc123xyz'.
First, I need to check which functions are available. The relevant ones here are probably related to the travel system since they mentioned budgeting and converting currency. Looking at the functions, there's '`set_budget_limit`' which requires an access token and a budget limit in USD. But the user provided the budget in RMB. So I need to convert 50,000 RMB to USD.
Wait, there's also '`compute_exchange_rate`' which can convert between currencies. The parameters for that function include `base_currency`, `target_currency`, and value. The user's budget is in RMB (`base_currency`: `RMB`), and they want it in USD (`target_currency`: `USD`). The value is 50,000. So I should first call `compute_exchange_rate` with these parameters to get the USD equivalent. Once I have the converted USD amount, I can then use '`set_budget_limit`' with the access token 'abc123xyz' and the converted USD value to set the budget. But since the user mentioned put a firm budget in place, maybe they just want the limit set in RMB? Wait, the function '`set_budget_limit`' specifies the budget limit in USD. So the user's 50,000 RMB needs to be converted to USD first.
So the steps are:
1. Use `compute_exchange_rate` to convert 50,000 RMB to USD.
2. Use the resulting USD value with `set_budget_limit`.
But the user might not have mentioned the exchange rate, so I need to make sure to call `compute_exchange_rate` first. Let me check the parameters again. The `compute_exchange_rate` requires `base_currency`, `target_currency`, and value. The user's base is RMB, target is USD, value is 50000. Then, the response will give the `exchanged_value` in USD. Then, use that value in `set_budget_limit`. So the function calls would be: First, `compute_exchange_rate(base_currency='RMB'`, `target_currency='USD'`, `value=50000)`. Then, `set_budget_limit(access_token='abc123xyz'`, `budget_limit=exchanged_value)`. But since the user's message doesn't mention the exchange rate, I need to perform the conversion first. Therefore, the correct sequence is to first call `compute_exchange_rate`, then use the result in `set_budget_limit`. However, the user might not have provided the exchange rate, so the assistant needs to handle that step.
`</think>`
`[compute_exchange_rate(base_currency='RMB', target_currency='USD', value=50000), set_budget_limit(access_token='abc123xyz', budget_limit=exchanged_value)]`}

---

---

**D-CORE-8B**

{**"role"**: "user", **"content"**: "Gearing up for some exciting travel adventures! I wish to put a firm budget in place, setting a cap equivalent to 50,000 RMB for my upcoming European excursions. I'd appreciate if you could take charge of the conversion and establish a budget framework for me using the access token 'abc123xyz'."}

{**"role"**: "assistant", **"content"**:" `<think>` To complete this task, I need to break it down into the following subtasks:
1. Use the '`compute_exchange_rate`' tool to convert 50,000 RMB to USD for budget setting, specifying 'RMB' as base currency and 'USD' as target currency.
2. Use the '`set_budget_limit`' tool to establish the converted USD amount as the budget framework using the provided access token 'abc123xyz'.
Now I should analyze the execution process of subtask 1. Okay, let's see. The user wants to convert 50,000 RMB to USD using the `compute_exchange_rate` tool. First, I need to check the parameters required for that function. The tool needs `base_currency`, `target_currency`, and value. The user specified RMB as the base and USD as the target, with a value of 50,000. So I should call `compute_exchange_rate` with those parameters. Let me make sure the parameters are correctly formatted. `Base_currency` is 'RMB', `target_currency` is 'USD', and value is 50000. That should do it. No other functions are needed here since the question is straightforward. `</think>` `[compute_exchange_rate(base_currency=\"RMB\", target_currency=\"USD\", value=50000)]`

{**"role"**: "tool", **"content"** : "`exchanged_value`": 7142.86}

{**"role"**: "assistant", **"content"**:" `<think>` Now I should analyze the execution process of subtask 2:Use the '`set_budget_limit`' tool to establish the converted USD amount as the budget limit using access token 'abc123xyz'.
Okay, the user wants to set a budget limit using the converted USD amount from the previous exchange rate calculation. The exchange rate tool returned 7142.86 USD for 50,000 RMB. Now, they need to use the '`set_budget_limit`' function with the access token 'abc123xyz' and the budget limit as 7142.86.
First, I check the parameters required for '`set_budget_limit`'. The required parameters are '`access_token`' and '`budget_limit`'. The user provided the access token 'abc123xyz' and the converted value 7142.86.
I need to make sure the budget limit is a float, which it is. So the function call should be `set_budget_limit(access_token='abc123xyz', budget_limit=7142.86)`.
No other parameters are needed, and the function should execute correctly. The response will confirm the budget limit set in USD. `</think>`
`[set_budget_limit(access_token=\"abc123xyz\", budget_limit=7142.86)]`}

---

A.7.2 CASE STUDY ON $\tau$-BENCH AIRLINE 6

---

**Qwen3-8B**

{**"role"**: "user",**"content"**: "My user ID is `aarav_garcia_1177`. I don't have the reservation ID right now, but I want to change my flight to the cheapest economy option for the day after my original booking. I'm okay with the original payment being refunded as credit."},

{**"role"**: "assistant", **"content"**: "[`get_user_details`]"},

{**"role"**: "tool", **"content"**: "`user_details`"},

{**"role"**: "assistant", **"content"**: "`<think>` Okay, let's see. The user wants to change their flight from ATL to PHL. They mentioned they don't have the reservation ID but want the cheapest economy option the day after their original booking. They're okay with the original payment being refunded as credit.

First, I need to get the user's reservations. The user provided their user ID, so I called `get_user_details`. The response shows they have two reservations: M05KNL and UHDAHF. Since they don't have the reservation ID, I should check both or ask for more details. But the user mentioned the day after the original booking. Wait, the original booking's date isn't provided yet. Maybe I need to find out the original flight dates.

Wait, the user's original flight was from ATL to PHL. But without the reservation ID, I can't get the specific flight details. The user might not know the reservation ID, so maybe I should proceed with the available reservations. However, the system requires the reservation ID to modify a flight. Since the user doesn't have it, I need to handle this.

Looking at the tools, there's a function to `get_reservation_details`, but that requires the reservation ID. Since the user doesn't have it, I can't proceed with modifying the flight yet. The user might need to provide the reservation ID, but they mentioned they don't have it. Alternatively, maybe the system can find the relevant reservation based on other info, but the tools don't support that.

Wait, the user is asking to change the flight to the cheapest economy option the day after the original booking. Maybe I can first find the original flight's date. But without the reservation ID, I can't get the original flight details. So, the next step is to inform the user that I need the reservation ID to proceed. However, the user might not have it.

Alternatively, maybe the user's original booking is the most recent one. Let me check the reservations. The user has two reservations: M05KNL and UHDAHF. The current date is 2024-05-15. If the original booking was made recently, maybe one of these reservations is the one they're referring to.

But without knowing the dates, I can't be sure.

Since the user is a gold member, they have 3 free checked bags for economy. But that's for baggage, not flight changes. The user wants to change the flight to the cheapest economy option the day after. So, perhaps I need to first find the original flight's date, then search for flights the next day. But without the reservation ID, I can't get the original flight's date.

Therefore, the correct step is to inform the user that I need the reservation ID to proceed with modifying their flight. Since they don't have it, maybe they need to check their account or contact support. Alternatively, if the system can retrieve the reservation based on other info, but the tools don't allow that. So, the next step is to ask the user to provide the reservation ID or transfer to a human agent. But according to the policy, transfer is only if the request can't be handled. Since the user is asking for a modification, which requires the reservation ID, and they don't have it, the agent can't proceed. Therefore, the correct action is to transfer to a human agent with the summary that the user needs the reservation ID to modify their flight.
`</think>`
[`transfer_to_human_agents`(summary=Üser wants to change their flight from ATL to PHL but does not have the reservation ID. They mentioned wanting the cheapest economy option the day after their original booking and are okay with refunding the original payment as credit. The agent needs the reservation ID to proceed with the modification.)]"}

---

---

**D-CORE-8B**

{**"role"**: "user", **"content"**: "My user ID is `aarav_garcia_1177`. I don't have the reservation ID right now, but I want to change my flight to the cheapest economy option for the day after my original booking. I'm okay with the original payment being refunded as credit."},

{**"role"**: "assistant", **"content"**: "[`get_user_details`]"},

{**"role"**: "tool", **"content"**: "`user_details`"},

{**"role"**: "assistant", **"content"**: "`<think>` OK, let me analyze the overall task: My user ID is `aarav_garcia_1177`. I don't have the reservation ID right now, but I want to change my flight to the day after the original one, and I'm looking for the cheapest economy option.
To complete this task, I need to break it down into the following subtasks:
1. Retrieve user details for `user_id` 'aarav_garcia_1177' to identify all reservations associated with the user.
2. Retrieve details for reservation 'M05KNL' to check if it matches the ATL-PHL route and verify flight details.
3. Retrieve details for reservation 'UHDAHF' to check if it matches the ATL-PHL route and verify flight details.
I have already solve the subtask:
1.Retrieve user details for `user_id` 'aarav_garcia_1177' to identify all reservations associated with the user.
Now I should analyze the execution process of subtask 2:Retrieve details for reservation 'M05KNL' to check if it matches the ATL-PHL route and verify flight details.
Okay, the user wants to change their flight from ATL to PHL. They provided their user ID, so I first retrieved their details. The user has two reservations: M05KNL and UHDAHF. Now, I need to check which of these reservations is for the ATL-PHL route.
I should start by getting the details for reservation M05KNL. Let me call `get_reservation_details` with `reservation_id` 'M05KNL'. Once I have that information, I can check the origin and destination of the flights in that reservation. If it's ATL to PHL, then I can proceed to help them modify the flight. If not, I'll need to check the other reservation, UHDAHF.
Assuming the reservation details come back, I'll look at the flights array. Each flight object should have origin and destination. If the origin is ATL and destination is PHL, then that's the one. If there are multiple flights, I need to confirm which segment they want to change. Also, I need to check the status of the flight to ensure it's still available for modification. If the flight is already departed, the user might need a different solution.
Once confirmed, I can proceed to update the reservation. Since the user wants to change the flight, I'll need to use `update_reservation_flights`. But first, I must make sure the new flight is available. I might need to search for available flights on the desired date. However, the user hasn't specified a new date yet. Wait, in the initial message, the user just said they need to change the flight but didn't mention the new date. So maybe I should ask for the new date after confirming the current reservation.
Wait, the user's original request was to change the flight from ATL to PHL. They might have a specific date in mind. But since they didn't provide it, perhaps I need to prompt them for the new date once I confirm the existing reservation. Alternatively, maybe the reservation M05KNL has a specific date that I can check. Let me proceed step by step.
First, get the reservation details for M05KNL. Then, check if the flights in that reservation are ATL to PHL. If yes, then I can ask the user for the new date they want to change to. Then, search for available flights on that date and proceed with the update. If the reservation doesn't match ATL-PHL, check UHDAHF next. `</think>`
[`get_reservation_details(reservation_id=M05KNL)`"} ]

---

### A.7.3 CASE STUDY ON $\tau$-BENCH AIRLINE 30

---

**Qwen3-8B**

{**"role"**: "user", **"content"**: "I think there might be an issue with my flight bookings. I accidentally ended up with multiple flights on the same days. Can you help me check if thať2019s actually been booked under my profile?"},

{**"role"**: "assistant", **"content"**: "[get_user_details]"},

{**"role"**: "tool", **"content"**: "user_details"},

{**"role"**: "assistant", **"content"**: "<think> Okay, let's see. The user is asking to check all their reservations for overlapping flights. They have seven reservations listed. My task is to check each reservation's flights to see if there are any flights on the same day.
First, the function get_reservation_details requires a reservation_id. Since there are seven reservations, I'll need to call this function seven times, once for each reservation ID.
Wait, but the user might not know which ones are problematic. So I should check all of them. Let me start with the first reservation, MFRB94. I'll call get_reservation_details with that ID. Then proceed to the next one, PUNERT, and so on until FDZ0T5, HTR26G, and 5BGGWZ.
Each time I get the reservation details, I'll look at the flights array. For each flight, note the date and check if there are multiple flights on the same date. If any reservation has flights on the same day, I'll need to inform the user about that. Also, I should check if the flights are on the same day across different reservations, but the user mentioned m̈ultiple flights on the same days,̈which might refer to within the same reservation or across different ones. The initial problem statement says m̈ultiple flights on the same days,̈so maybe they have two separate flights on the same day in different reservations. But the user is asking to check all reservations for any overlaps.
Wait, the user's original message was about m̈ultiple flights on the same days,̈which could mean that within a single reservation, there are two flights on the same day, which isn't possible for a round trip. Or maybe they have two separate reservations with flights on the same day. But the user is asking to check all their reservations to see if any flights are on the same day. So I need to check each reservation's flights for same-day flights and also check across reservations if there's any overlap.
However, the user might be referring to within their own bookings, not overlapping with others. But the problem is that the user might have two separate reservations with flights on the same day, which could be an issue if they're planning trips.
But according to the policy, the agent can only check the reservations under the user's profile. So I need to check each reservation's flights for same-day flights. Let me start by getting the details for each reservation one by one. Once I have all the flight details, I can compile them and check for any duplicates or overlaps.
So the next step is to call get_reservation_details for each reservation ID provided. Let's start with the first one, MFRB94. </think>
[get_reservation_details(reservation_id=\"MFRB94\")]"}

---

---

**D-CORE-8B**

{**"role"**: "user", **"content"**: "I think there might be an issue with my flight bookings. I accidentally ended up with multiple flights on the same days. Can you help me check if that2019s actually been booked under my profile?"},

{**"role"**: "assistant", **"content"**: "[get_user_details]"},

{**"role"**: "tool", **"content"**: "user_details"},

{**"role"**: "assistant", **"content"**: "<think> OK, let me analyze the overall task: Sure, my user ID is sophia_martin_4574.
To complete this task, I need to break it down into the following subtasks:
1. Retrieve user details for user_id 'sophia_martin_4574' to identify all reservations associated with the user.
2. Fetch reservation details for reservation_id 'MFRB94' to check flight bookings for the same day.
3. Fetch reservation details for reservation_id 'PUNERT' to check flight bookings for the same day.
4. Fetch reservation details for reservation_id 'HSR97W' to check flight bookings for the same day.
5. Fetch reservation details for reservation_id 'SE9KEL' to check flight bookings for the same day.
6. Fetch reservation details for reservation_id 'FDZ0T5' to check flight bookings for the same day.
7. Fetch reservation details for reservation_id 'HTR26G' to check flight bookings for the same day.
I have already solve the subtask:
1.Retrieve user details for user_id 'sophia_martin_4574' to identify all reservations associated with the user.
Now I should analyze the execution process of subtask 2:Fetch reservation details for reservation_id 'MFRB94' to check flight bookings for the same day.
Okay, the user wants to check if there are multiple flights booked for the same day. They provided their user ID, and I retrieved their reservations. Now, I need to look into each reservation to see the flight dates.
First, I'll start with the first reservation, 'MFRB94'. I'll call get_reservation_details for that ID. Once I have the details, I'll check the flights' dates. If there are multiple flights on the same date, I'll note that. Then, I'll proceed to the next reservation IDs listed: 'PUNERT', 'HSR97W', and so on. For each reservation, I'll repeat the process of fetching details and checking the flight dates. I need to make sure I don't miss any reservations. The user mentioned a mix-up with their assistant, so it's possible that multiple bookings were made accidentally. By checking each reservation's flight dates, I can identify if any dates are duplicated. Once I have all the information, I can inform the user which reservations have conflicting dates and offer to help cancel or modify any duplicate bookings. </think>
[get_reservation_details(reservation_id=\"MFRB94\")"} ]

## A.8 REWARD FUNCTION IN DA-GRPO

The reward employed in DA-GRPO aligns with that of ToolRL (Qian et al., 2025):

$$R_{\text{format}} = \mathbb{1}\left(\texttt{"<think>\textbackslash n".*"\textbackslash n</think>\textbackslash n\textbackslash n"}\right), \quad R_{\text{struct}} = \mathbb{1}(\mathcal{N}_G = \mathcal{N}_P), \qquad (10)$$

$$R_{\text{key}} = \frac{1}{|\mathcal{K}|}\sum_{j=1}^{|\mathcal{K}|}\mathbb{1}(\mathcal{K}_j^G = \mathcal{K}_j^P), \quad R_{\text{value}} = \frac{1}{|\mathcal{K}|}\sum_{j=1}^{|\mathcal{K}|}\frac{1}{|\mathcal{V}_j|}\sum_{k=1}^{|\mathcal{V}_j|}\mathbb{1}(\mathcal{V}_j^G[k] = \mathcal{V}_j^P[k]), \qquad (11)$$

$$R_i = R_{\text{format}} + R_{\text{struct}} + R_{\text{key}} + R_{\text{value}}, \qquad (12)$$

where the $\mathbb{1}$ stands for exact matching. $\mathcal{N}$ are tool names, $\mathcal{K}$ are parameter names, $\mathcal{V}$ are parameter values, $P$ and $G$ represents predicted and ground truth.

## A.9  Proof of DA-GRPO

The objective of DA-GRPO:

$$\mathcal{J}_{\text{DA-GRPO}}(\theta) = \mathbb{E}[\min(\frac{\pi_\theta(y_{i,t} \mid x_i, y_{i,<t})}{\pi_{\text{ref}}(y_{i,t} \mid x_i, y_{i,<t})}\hat{A}_{i,t}, \text{clip}(\frac{\pi_\theta(y_{i,t} \mid x_i, y_{i,<t})}{\pi_{\text{ref}}(y_{i,t} \mid x_i, y_{i,<t})}, 1 - \epsilon, 1 + \epsilon)\hat{A}_{i,t}) \tag{13}$$
$$- \lambda\mathbb{D}_{\text{KL}}[\pi_\theta||\pi_{\text{ref}}]],$$

where the advantage function as:

$$\hat{A}_{i,t} = \begin{cases} A_{i,t}, & \text{if } A_{i,t} \neq 0, \\ \psi(\mathcal{H}_{i,t}), & \text{if } A_{i,t} = 0, \end{cases} \tag{14}$$

where $\mathcal{H}_{i,t} = -\sum_v \pi_\theta(v|x_i, y_{i,<t}) \log \pi_\theta(v|x_i, y_{i,<t})$ is the entropy of the policy distribution at position $t$. We then define importance sampling ratio $r_{i,t}(\theta) = (\frac{\pi_\theta(y_{i,t}|x_i,y_{i,<t})}{\pi_{\text{ref}}(y_{i,t}|x_i,y_{i,<t})})$, the objective of DA-GRPO becomes:

$$\mathcal{J}_{\text{DA-GRPO}}(\theta) = \mathbb{E}[\min(r_{i,t}(\theta)\hat{A}_{i,t}, \text{clip}(r_{i,t}(\theta), 1 - \epsilon, 1 + \epsilon)\hat{A}_{i,t}) - \lambda\mathbb{D}_{\text{KL}}[\pi_\theta||\pi_{\text{ref}}]]. \tag{15}$$

After omitting the KL term, the policy gradient of DA-GRPO can be formulated as:

$$\nabla_\theta \mathcal{J}_{\text{DA-GRPO}} = \mathbb{E}[r_{i,t}(\theta)\hat{A}_{i,t}\nabla_\theta \log \pi_\theta(y_{i,t}|x_i, y_{i,<t})] \tag{16}$$

Define two disjoint index sets:

$$\mathcal{T}_{\neq 0} = \{(i,t) : \hat{A}_{i,t} \neq 0\}, \tag{17}$$
$$\mathcal{T}_{=0} = \{(i,t) : \hat{A}_{i,t} = 0\}. \tag{18}$$

Then the policy gradient decomposes as:

$$\nabla_\theta \mathcal{J}_{\text{DA-GRPO}} = \underbrace{\frac{1}{G} \sum_{(i,t) \in \mathcal{T}_{\neq 0}} \frac{1}{|y_i|} r_{i,t}(\theta) A_{i,t} \nabla_\theta \log \pi_\theta(y_{i,t}|x_i, y_{i,<t})}_{\text{Original GRPO Term}}$$
$$+ \underbrace{\frac{1}{G} \sum_{(i,t) \in \mathcal{T}_{=0}} \frac{1}{|y_i|} r_{i,t}(\theta) \psi(\mathcal{H}_{i,t}) \nabla_\theta \log \pi_\theta(y_{i,t}|x_i, y_{i,<t})}_{\text{Entropy Advantage Term}} \tag{19}$$

**Theorem 1** (Prevention of Learning Stagnation). *Let $\mathcal{T}_{=0} \neq \emptyset$ be the set of positions where $A_{i,t} = 0$. If there exists $(i,t) \in \mathcal{T}_{=0}$ such that:*

*1. $r_{i,t}(\theta) \neq 0$*

*2. $\psi(\mathcal{H}_{i,t}) > 0$ (i.e., $\pi_\theta(\cdot|q, o_{i,<t})$ is not degenerate)*

*Then $\|\nabla_\theta \mathcal{J}_{\text{DA-GRPO}}\| > 0$, ensuring continued learning.*

*Proof.* Assume all $(i,t)$ satisfy $A_{i,t} = 0$, so $\mathcal{T}_{\neq 0} = \emptyset$ and $\mathcal{T}_{=0} = \{(i,t) : 1 \leq i \leq G, 1 \leq t \leq |o_i|\}$.

The modified gradient becomes:

$$\nabla_\theta \mathcal{J}_{\text{DA-GRPO}} = \frac{1}{G} \sum_{(i,t) \in \mathcal{T}_{=0}} \frac{1}{|y_i|} r_{i,t}(\theta) \psi(\mathcal{H}_{i,t}) \nabla_\theta \log \pi_\theta(y_{i,t}|x_i, y_{i,<t}) \tag{20}$$

By the fundamental property of entropy:

$$\mathcal{H}_{i,t} = - \sum_a \pi_\theta(a|\cdot) \log \pi_\theta(a|\cdot) \geq 0 \tag{21}$$

with equality if and only if $\exists a^* : \pi_\theta(a^*|q, o_{i,<t}) = 1$ (degenerate distribution).

Consider any $(i,t) \in \mathcal{T}_{=0}$ satisfying conditions (1) and (2). The gradient contribution from this term is:

$$g_{i,t} = \frac{1}{G|o_i|} r_{i,t}(\theta) \psi(\mathcal{H}_{i,t}) \nabla_\theta \log \pi_\theta(o_{i,t}|q, o_{i,<t}) \tag{22}$$

Since:

- $r_{i,t}(\theta) \neq 0$ (by condition 1)

- $\psi(\mathcal{H}_{i,t}) > 0$ (by condition 2)

- $\nabla_\theta \log \pi_\theta(o_{i,t}|q, o_{i,<t}) \neq 0$ for non-degenerate distributions

We have $\|g_{i,t}\| > 0$, which implies:

$$\|\nabla_\theta \mathcal{J}_{\text{new}}\| \geq \|g_{i,t}\| > 0 \tag{23}$$

Therefore, the gradient is non-zero and learning continues. $\square$

**Theorem 2** (Entropy Reduction Property of DA-GRPO). *When $A_{i,t} = 0$ for some token position $(i,t)$, DA-GRPO encourages the generation of high-entropy tokens by reducing their entropy. Specifically, for $(i,t) \in \mathcal{T}_{=0}$, the gradient contribution is:*

$$\nabla_\theta \mathcal{J}_{i,t} = \frac{1}{G|y_i|} r_{i,t}(\theta) \psi(\mathcal{H}_{i,t}) \nabla_\theta \log \pi_\theta(y_{i,t}|x_i, y_{i,<t}) \tag{24}$$

*where $\mathcal{H}_{i,t} = - \log \pi_{\theta_{old}}(y_{i,t}|x_i, y_{i,<t})$ is the detached cross-entropy. Since $r_{i,t}(\theta) > 0$ and tokens with higher $\mathcal{H}_{i,t}$ (lower probability under $\pi_{\theta_{old}}$) receive stronger positive gradients, DA-GRPO preferentially increases the probability of high-entropy tokens, thereby reducing their entropy and making them more likely to be generated.*

*Proof.* For any token position $(i,t) \in \mathcal{T}_{=0}$ where $A_{i,t} = 0$, the gradient contribution is:

$$\nabla_\theta \mathcal{J}_{i,t} = \frac{1}{G|y_i|} r_{i,t}(\theta) \psi(\mathcal{H}_{i,t}) \nabla_\theta \log \pi_\theta(y_{i,t}|x_i, y_{i,<t}) \tag{25}$$

**Key observations:**

1. The entropy term $\mathcal{H}_{i,t} = - \log \pi_{\theta_{old}}(y_{i,t}|x_i, y_{i,<t})$ is detached from the current policy $\pi_\theta$, acting as a fixed coefficient based on the old policy.

2. For tokens with low probability under $\pi_{\theta_{old}}$:
$$\pi_{\theta_{old}}(y_{i,t}|x_i, y_{i,<t}) \text{ small} \Rightarrow \mathcal{H}_{i,t} = - \log \pi_{\theta_{old}}(y_{i,t}|x_i, y_{i,<t}) \text{ large} \tag{26}$$

3. Since $r_{i,t}(\theta) > 0$ and $\mathcal{H}_{i,t} \geq 0$, the gradient always points in the direction of increasing $\log \pi_\theta(y_{i,t}|x_i, y_{i,<t})$.

4. **Crucially**, the magnitude of the gradient is proportional to $\mathcal{H}_{i,t}$: tokens that had higher entropy (lower probability) under $\pi_{\theta_{old}}$ receive stronger positive gradients.

**Entropy reduction mechanism:**

Consider two sampled tokens at positions where $A_{i,t} = 0$:

- Token $y_i^{(1)}$ with high entropy: $\pi_{\theta_{old}}(y_i^{(1)}|\cdot) = 0.1 \Rightarrow \mathcal{H}_i^{(1)} \approx 2.3$

- Token $y_i^{(2)}$ with low entropy: $\pi_{\theta_{old}}(y_i^{(2)}|\cdot) = 0.8 \Rightarrow \mathcal{H}_i^{(2)} \approx 0.22$

The gradient magnitude for the high-entropy token is approximately $10\times$ larger than that for the low-entropy token. Therefore, DA-GRPO preferentially increases the probability of high-entropy tokens, which:

1. **Increases their likelihood**: $\pi_\theta(y_{i,t}|\cdot)$ increases for tokens that were uncertain under $\pi_{\theta_{old}}$

2. **Reduces their entropy**: As $\pi_\theta(y_{i,t}|\cdot)$ increases, the local entropy $-\log \pi_\theta(y_{i,t}|\cdot)$ decreases

3. **Makes them more deterministic**: The model becomes more confident about generating these previously uncertain tokens

This mechanism differs from standard entropy regularization (which would increase overall distribution entropy). Instead, DA-GRPO selectively reduces the entropy of high-entropy tokens that were sampled, encouraging exploration while consolidating discovered behaviors. $\qquad\square$

**Remark 1.** *This entropy reduction property has important implications:*

- ***Exploration exploitation**: DA-GRPO encourages sampling diverse tokens (high-entropy) but then commits to them by reducing their entropy.*

- ***Stability**: Unlike additive entropy bonuses that can lead to unbounded entropy growth, DA-GRPO's mechanism is self-limiting—once a token's probability increases, its $\mathcal{H}_{i,t}$ in future iterations decreases.*

- ***Advantage-aware**: This mechanism only applies when $A_{i,t} = 0$, preserving the advantage signal where it matters while providing structured exploration elsewhere.*

## A.10   LLM USAGE

Large Language Models (LLMs) were used solely for linguistic assistance in manuscript preparation, including sentence rephrasing, grammar checking, and improving readability. The LLM had no involvement in research ideation, methodology, experimental design, or data analysis. All scientific content and analyses were developed by the authors, who take full responsibility for the manuscript content and ensure compliance with ethical guidelines.

