# OpenReview forum: "D-CORE: Incentivizing Task Decomposition in Large Reasoning Models for Complex Tool Use"
_ICLR.cc/2026/Conference — Submitted to ICLR 2026_

### Official Review · Reviewer_jzW5 · 2025-10-27

**Soundness:** 3
**Presentation:** 3
**Contribution:** 3
**Rating:** 4
**Confidence:** 3

**Summary:**

he paper identifies a critical failure mode called “Lazy Reasoning” in Large Reasoning Models (LRMs): when faced with complex tool-use or multi-turn reasoning tasks, LRMs tend to loop through meaningless reflective thoughts instead of decomposing tasks into subtasks.

To address this, the authors propose D-CORE, a two-stage training framework:

Self-Distillation Stage – teaches the LRM to perform task decomposition and subtask reasoning by generating and reusing its own reasoning trajectories.

Diversity-Aware RL (DA-GRPO) – reintroduces reflective diversity by adding entropy-based terms into the advantage function to encourage exploration without reverting to unproductive reasoning.

**Strengths:**

This paper presents a well-motivated and empirically grounded study on addressing the “Lazy Reasoning” phenomenon in large reasoning models (LRMs), offering a novel and systematic framework called D-CORE that combines self-distillation with diversity-aware reinforcement learning. The problem is clearly identified through quantitative and qualitative evidence, showing how excessive reflection hinders multi-turn tool use reasoning. The proposed two-stage method is conceptually coherent—self-distillation promotes task decomposition, while the entropy-shaped advantage in DA-GRPO effectively restores reflective diversity. The paper demonstrates strong empirical results across BFCLv3 and τ-Bench, with significant improvements over both open-source and proprietary baselines, and provides detailed behavioral analysis that connects reasoning structure to performance. The overall experimental design, ablation studies, and large-scale evaluations reflect thoughtful execution and a meaningful step forward in enhancing compositional reasoning for tool-use scenarios.

**Weaknesses:**

The entropy-based advantage formulation is only intuitively justified, without a formal connection to reasoning diversity or convergence guarantees. Several implementation details—such as the construction of the 40K self-distillation dataset, the specific prompt templates, and hyperparameter settings—are deferred to the appendix or omitted entirely, making the training pipeline hard to replicate. The evaluation scope is relatively narrow, relying only on BFCLv3 and τ-Bench, both focused on structured tool use, leaving questions about cross-domain generalization. Additionally, qualitative comparisons of reasoning trajectories before and after D-CORE are missing, which weakens claims about interpretability. Minor but pervasive grammatical inconsistencies and figure referencing issues also reduce readability and polish. Overall, while the framework is promising, the paper would benefit from deeper theoretical analysis, richer qualitative evidence, and clearer experimental documentation to meet the standards of a top-tier ICLR submission.  Please provide clarifications on the questions asked above.

**Questions:**

Please address the questions in the weakness

---

> ### Author Response · Authors · 2025-11-22
> **Response to Reviewer jzW5 (1/3)**
>
> We sincerely thank the reviewer `jzW5`  for taking your valuable time to evaluate our work.  We appreciate your feedback, which motivated us to provide additional analysis that strengthens our work. Below is our point-by-point response to your feedback:
>
> > **W1:The entropy-based advantage formulation is only intuitively justified, without a formal connection to reasoning diversity or convergence guarantees.**
>
> A1: Thank you for this insightful question. We have created detailed plots showing the evolution of reward, reflection token numbers, and entropy during DA-GRPO training on tool use dataset. These plots are included in the revised paper as Figure 7. The figure clearly shows that adding entropy-based advantages yields three effects: higher reward, more reflection tokens, and lower entropy. These trends match [1], confirming DA-GRPO's role in shaping RL training dynamics.
>
> [1] Cheng D, Huang S, Zhu X, et al. Reasoning with exploration: An entropy perspective[J]. arXiv preprint arXiv:2506.14758, 2025.
>
> > **W2: Several implementation details—such as the construction of the 40K self-distillation dataset, the specific prompt templates, and hyperparameter settings—are deferred to the appendix or omitted entirely, making the training pipeline hard to replicate.**
>
> A2: Thank you for raising this important concern. We would like to assure you that we are fully committed to open-sourcing our work to benefit the community:
>
> - Code Release: We will release the complete D-CORE data generation pipeline and model training code on GitHub.
>
> - Model Release: We will release both D-CORE-8B and D-CORE-14B models on Hugging Face.
>
> - Data Release: We will also release our training data on Hugging Face to ensure full reproducibility.
>
> We believe these releases will enable the community and researchers to fully follow and reproduce our work. Additionally, we have improved the appendix formatting so all prompts and code now appear on single pages for easier reference. We hope this addresses your concerns regarding reproducibility.
>
> >**W3: Minor but pervasive grammatical inconsistencies and figure referencing issues also reduce readability and polish.**
>
> A3: We have addressed the grammatical and figure citation issues in the submitted revised manuscript. We appreciate your valuable feedback.

---

> ### Author Response · Authors · 2025-11-22
> **Response to Reviewer jzW5 (2/3):**
>
> >**W4: Additionally, qualitative comparisons of reasoning trajectories before and after D-CORE are missing, which weakens claims about interpretability.**
>
> A4: We present one case on the current page, while additional qualitative comparisons have been included in the appendix. We sincerely appreciate your valuable feedback on our paper. The following case study compares the reasoning trajectories of Qwen3-8B and D-CORE-8B on Question 196 from BFCLv3-MultiTurn Base:
>
> D-CORE-8B immediately engages in task decomposition from the very beginning of its reasoning process. It recognizes that completing this query requires breaking it down into two multi-step subtasks. It then proceeds to consider how to accomplish the first subtask. After completing the first task and executing the corresponding function, D-CORE-8B moves on to reasoning about the second subtask, ultimately arriving at the correct answer.
>
> Query:
> ```
> User: "Gearing up for some exciting travel adventures! I wish to put a firm budget in place, setting a cap equivalent to 50,000 RMB for my upcoming European excursions. I'd appreciate if you could take charge of the conversion and establish a budget framework for me using the access token 'abc123xyz'."
> ```
> Qwen3-8B:
> ```
> Assistant: "<think>Okay, let's see. The user wants to set a budget limit of 50,000 RMB for their European travels and needs the conversion handled using the access token 'abc123xyz'.
> First, I need to check which functions are available. The relevant ones here are probably related to the travel system since they mentioned budgeting and converting currency. Looking at the functions, there's 'set_budget_limit' which requires an access token and a budget limit in USD.
>
> But the user provided the budget in RMB. So I need to convert 50,000 RMB to USD.
>
> Wait, there's also 'compute_exchange_rate' which can convert between currencies. The parameters for that function include base_currency, target_currency, and value. The user's budget is in RMB (base_currency: RMB), and they want it in USD (target_currency: USD). The value is 50,000. So I should first call compute_exchange_rate with these parameters to get the USD equivalent. Once I have the converted USD amount, I can then use 'set_budget_limit' with the access token 'abc123xyz' and the converted USD value to set the budget.
>
> But since the user mentioned \"put a firm budget in place\", maybe they just want the limit set in RMB?
>
> Wait, the function 'set_budget_limit' specifies the budget limit in USD. So the user's 50,000 RMB needs to be converted to USD first.
> So the steps are:
>  1. Use compute_exchange_rate to convert 50,000 RMB to USD.
>  2. Use the resulting USD value with set_budget_limit.
>
> But the user might not have mentioned the exchange rate, so I need to make sure to call compute_exchange_rate first. Let me check the parameters again. The compute_exchange_rate requires base_currency, target_currency, and value. The user's base is RMB, target is USD, value is 50000. Then, the response will give the exchanged_value in USD. Then, use that value in set_budget_limit.
>
> So the function calls would be:
> First, compute_exchange_rate(base_currency='RMB', target_currency='USD', value=50000). Then, set_budget_limit(access_token='abc123xyz', budget_limit=exchanged_value).
> </think>
> [compute_exchange_rate(base_currency='RMB', target_currency='USD', value=50000), set_budget_limit(access_token='abc123xyz', budget_limit=exchanged_value)]"
> ```
> D-CORE-8B:
> ```
> Assistant: "<think> To complete this task, I need to break it down into the following subtasks:
>
> 1. Use the 'compute_exchange_rate' tool to convert 50,000 RMB to USD for budget setting, specifying 'RMB' as base currency and 'USD' as target currency.
>
> 2. Use the 'set_budget_limit' tool to establish the converted USD amount as the budget framework using the provided access token 'abc123xyz'.
>
> Now I should analyze the execution process of subtask 1. Okay, let's see. The user wants to convert 50,000 RMB to USD using the compute_exchange_rate tool.
>
> First, I need to check the parameters required for that function. The tool needs base_currency, target_currency, and value. The user specified RMB as the base and USD as the target, with a value of 50,000. So I should call compute_exchange_rate with those parameters.
>
> Let me make sure the parameters are correctly formatted. Base_currency is 'RMB', target_currency is 'USD', and value is 50000. That should do it. No other functions are needed here since the question is straightforward.
> </think>
>  [compute_exchange_rate(base_currency=\"RMB\", target_currency=\"USD\", value=50000)]"
> Tool: {"exchanged_value": 7142.86}
> Assistant: "<think> Now I should analyze the execution process of subtask 2:Use the 'set_budget_limit' tool to establish the converted USD amount as the budget limit using access token 'abc123xyz'.
> ...
> </think>
> [set_budget_limit(access_token=\"abc123xyz\", budget_limit=7142.86)]"
> ```

---

> ### Author Response · Authors · 2025-11-22
> **Response to Reviewer jzW5 (3/3)**
>
> > **W5: The evaluation scope is relatively narrow, relying only on BFCLv3 and τ-Bench, both focused on structured tool use, leaving questions about cross-domain generalization.**
>
> A5: Thank you very much for raising this important concern. Since you and reviewer `dxrp` share the same concern, we have prepared a unified response to address both of your questions. To thoroughly address your concerns about generalization, we have conducted three additional sets of experiments on  $\tau^2$-Bench, ACEBench, and BFCLv4-Agentic :
> -  $\tau^2$-Bench is a novel benchmark for evaluating conversational agents in dual-control environments, where both the user and the agent can perform actions to affect the shared world state.  Our experimental results demonstrate that D-CORE maintains strong performance even under these more demanding conditions, particularly on the newly introduced telecom tasks, which validates the effectiveness of our approach.
> | Model | Retail &uarr;| Airline &uarr;| Telecom&uarr; | Avg &uarr;|
> |:------|:-------|:--------|:--------|:----|
> | Qwen3-8B | 41.5 | 31.3 | 26.3 | 33.0 |
> | Qwen3-14B | 46.5 | 30.0 | 31.7 | 36.1 |
> | Qwen3-32B | 49.1 | 36.0 | 28.4 | 37.8 |
> | xLAM2-8B | 51.3 | 35.4 | 22.4 | 36.4 |
> | xLAM2-32B | 53.1 | 41.6 | 26.1 | 40.3 |
> | xLAM2-70B | **54.9** | **46.0** | 29.3 | 43.4 |
> | D-CORE-8B | 49.8 | 38.4 | 30.5 | 39.6 |
> | D-CORE-14B | 53.5 | 44.1 | **34.9** | **44.2** |
>
> - ACEBench employs more complex system and user prompts, revealing that fine-tuned LLMs that perform well on BFCLv3 actually underperform compared to open-source general-purpose models on this benchmark. This confirms that SFT-trained LLMs suffer from degraded generalization capabilities. The experimental results on ACEBench demonstrate that our model achieves further improvements over the Qwen3 baseline.
> | Model | Normal &uarr;| Special &uarr;| Agent &uarr;| Overall &uarr;|
> |:------|:------|:-------|:-----|:-------|
> | Qwen3-8B | 71.4 | 75.3 | 29.1 | 65.9 |
> | Qwen3-14B | 66.9 | 84.0 | 44.2 | 68.0 |
> | Qwen3-32B | 75.9 | 77.3 | 49.2 | 72.2 |
> | xLAM2-8B | 58.8 | 0.0 | 5.0 | 34.8 |
> | xLAM2-32B | 69.2 | 24.7 | 13.4 | 52.5 |
> | xLAM2-70B | 57.1 | 5.3 | 38.4 | 36.5 |
> | D-CORE-8B | **77.9** | 78.7 | **59.2** | 75.2 |
> | D-CORE-14B | 77.1 | **88.7** | 55.8 | **76.9** |
>
> - BFCLv4-agentic introduces two innovative evaluation scenarios:1.Web-search task: Tests multi-hop question answering capabilities using web search tools. 2.Memory task: Evaluates conversational memory management using dedicated memory tools.
> | Model | Memory KV &uarr;|Memory Vector &uarr; |Memory  Recursive Sum &uarr; |  Web Search Base &uarr; |
> |-------|---------------|-------------------|-------------|---------------|
> | Qwen3-8B | 14.8 | 14.8 | 27.1 | 16.0 |
> | Qwen3-14B | 15.5 | 17.4 | 42.6 | 34.0 |
> | Qwen3-32B | **15.5** | 20.7 | 38.7 | 32.0 |
> | xLAM2-8B | 4.5 | 14.2 | 28.4 | 8.0 |
> | xLAM2-32B | 8.4 | 10.4 | 31.6 | 31.0 |
> | xLAM2-70B | 5.2 | 11.6 | 36.1 | 13.0 |
> | D-CORE-8B | 14.0 | 17.0 | 23.0 | **36.0** |
> | D-CORE-14B | 11.6 | **24.9** | **42.9** | **39.0** |
>
> D-CORE achieves substantial improvements on the web-search task while maintaining baseline performance on the memory task.  After conducting a thorough analysis, we identified a strong correlation between the multi-hop nature of web search queries and our task decomposition approach:
>
> ```
> user: Many of the world's largest waterfalls are surrounded by vast jungles that contribute to their ecological richness. Which jungle surrounds the national park that has the largest waterfall in the country with the highest GDP growth in 2024?
> ```
>
> ```
> D-CORE-8B: <think> To complete this task, I need to break it down into the following subtasks:
>
> 1. Find the country with the highest GDP growth in 2024.
>
> 2. Find the largest waterfall in that country.
>
> 3. Find the national park that contains this waterfall.
>
> 4. Identify what jungle surrounds this national park.
>
> Now I should analyze the execution process of subtask 1: Find the country with the highest GDP growth in 2024. Let me start by searching for the country with the highest GDP growth in 2024. <\think>
> [search_engine_query(keywords='country highest GDP growth 2024')]
> ```
> These experimental results demonstrate that the decomposition capabilities learned from complex tool use successfully transfer to the new domain.
>
> We sincerely thank the reviewer `jzW5` for the constructive feedback. We have carefully addressed each concern through additional experiments, enhanced analysis, and comprehensive revisions. We believe these improvements have significantly strengthened both the technical contribution and presentation quality of our work. We are committed to further refinements if any concerns remain.

---

> ### Author Response · Authors · 2025-11-25
> **Seeking Feedback on Our Rebuttal Responses**
>
> Dear Reviewer `jzW5`,
>
> Thank you for your constructive feedback and for highlighting the areas where our paper could be strengthened.
>
> In our rebuttal, we have specifically addressed your concerns regarding:
> - **Deeper theoretical analysis**: We provided the training dynamics of DA-GRPO and a detailed explanation of DA-GRPO's motivation in the revised paper.
> - **Richer qualitative evidence**: We added three qualitative case studies of D-CORE-8B in the appendix, which clearly demonstrate the effectiveness of D-CORE.
> - **Clearer experimental documentation**: We provide additional results demonstrating D-CORE's performance on three new tasks.
>
> We have carefully answered all the questions you raised and believe our responses have adequately addressed the points needed to meet the standards you mentioned.
>
> We would greatly appreciate it if you could review our detailed responses when you have a chance. If any clarifications remain insufficient or if you have further questions, we are happy to provide additional information.
>
> Thank you for your time and valuable guidance.
>
> Best regards,
> Authors

---

> ### Author Response · Authors · 2025-11-26
> **Supplementary Experiments on DA-GRPO**
>
> Dear Reviewer `jzW5`,
> to further address your concerns regarding DA-GRPO, we have conducted an additional experiment to validate its effectiveness.
>
> In the revised manuscript, we have supplemented the motivation for DA-GRPO. The advantage $A_{i,t}$ in GRPO is defined as: $A_{i,t} = \frac{R_i - mean({R_i})}{std({R_i})}$. When the standard deviation $std({R_i})$ approaches zero, the advantage term $A_{i,t}$ vanishes, thereby eliminating the gradient signal.  This motivates us to seek an alternative formulation. Recent work [1][2] shows that high-entropy tokens correlate with reflective reasoning, which we confirm in Figure 6(b), and that entropy-based advantages promote reasoning diversity [1]. This leads to our motivation: **can we leverage an entropy-based advantage to address gradient vanishing when std approaches zero, thus recovering reasoning diversity?**
>
> The key idea is to scale each token’s gradient update in proportion to its detached entropy $\mathcal{H}^{\mathrm{detach}}$ when std approaches zero:
>
> $\mathcal{H}^{\mathrm{detach}}=-\sum_{v\in V}\pi_\theta(v|q,o_{<t})\log\pi_\theta(v|q,o_{<t}).$
>
> The entropy term $\mathcal{H}^{\mathrm{detach}}_{i,t}$ is detached during backpropogation, serving as a fixed offset that adjust update magnitude without affecting gradients. We then define an entropy-based advantage term $\psi(\mathcal{H}_t)$ to incentivize the generation of high-entropy tokens:
>
> $\psi(\mathcal{H}_t)=\min({\alpha} \cdot {{H}^{{detach}}},~\delta)$.
>
> Then we define the advantage function in DA-GRPO as:
>
> $\hat{A}_{i,t} = \psi (\mathcal{H_t})$ when $std({R_i})=0$, otherwise, it equals the original advantage.
>
> To validate the effectiveness of DA-GRPO, we conducted experiments using the dynamic sampling method proposed in DAPO [3]. Specifically, we over-sample and filter out prompts with $std({R_i})=0$, retaining only prompts with effective gradients in each batch while maintaining a consistent batch size. The results show that simply discarding samples with zero standard deviation yields suboptimal performance. For evaluation, we selected $\tau$-bench, where DA-GRPO achieves the most substantial improvements, along with the out-of-distribution $\tau^2$-bench dataset, to clearly demonstrate the efficacy of DA-GRPO:
>
> | Method || $\tau$-bench | || $\tau^2$-bench ||| Avg |
> |:--------:|:---------:|:---------:|:---------:|:--------:|:-------:|:-------:|:-------:|:--------:|
> | | Retail &uarr; | Airline &uarr;| Overall &uarr;| Retail &uarr;| Airline &uarr;| Telecom &uarr;| Overall &uarr;| (%) &uarr;|
> | GRPO | 49.2 | 41.2 | 45.2 | **50.2** | **40.1** | **34.0** | **41.4** | 43.3 |
> | GRPO + Dynamic Sampling | 56.8 | 38.8 | 47.8 | 49.5 | 35.3 | 27.0 | 37.3 | 42.6 |
> | DA-GRPO α=0.1 | **59.8** | **44.8** | **52.3** | 49.8 | 38.4 | 30.5 | 39.6 | **45.9** |
>
> The experimental results confirm that while dynamic sampling improves performance on $\tau$-bench by discarding samples with zero standard deviation, it degrades performance on the out-of-distribution $\tau^2$-bench. In contrast, DA-GRPO, which computes token entropy advantages for samples with zero standard deviation, achieves the most balanced performance across both benchmarks.
>
> We hope this experiment alleviates your concerns about DA-GRPO and welcome your feedback.
>
> [1] Cheng, Daixuan, et al. "Reasoning with exploration: An entropy perspective." arXiv preprint arXiv:2506.14758 (2025).
>
> [2] Wang, Shenzhi, et al. "Beyond the 80/20 rule: High-entropy minority tokens drive effective reinforcement learning for llm reasoning." arXiv preprint arXiv:2506.01939 (2025).
>
> [3] Yu, Qiying, et al. "Dapo: An open-source llm reinforcement learning system at scale." arXiv preprint arXiv:2503.14476 (2025).

---

### Official Review · Reviewer_q2LN · 2025-10-28

**Soundness:** 4
**Presentation:** 2
**Contribution:** 4
**Rating:** 8
**Confidence:** 3

**Summary:**

This paper introduces D-CORE (Decomposing tasks and Composing Reasoning processes), a two-stage training framework designed to enhance large reasoning models (LRMs) in complex multi-turn tool-use scenarios. The authors first identify a phenomenon termed Lazy Reasoning, where LRMs engage in lengthy but unproductive reflections instead of performing effective task decomposition. To address this issue, D-CORE first employs self-distillation to teach the model to decompose complex tasks into subtasks and compose coherent reasoning trajectories, and then applies diversity-aware reinforcement learning (DA-GRPO) to restore reflective and exploratory reasoning behaviors. Experimental results demonstrate clear performance improvements, and the paper includes comprehensive ablation studies to validate the contribution of each proposed component.

**Strengths:**

- The problem studied in this paper is interesting. The Lazy Reasoning phenomenon is indeed commonly observed when using large language models, making the motivation of this work clear and compelling.

- The paper proposes a novel and well-structured framework. The framework itself, along with the specific techniques employed in each component (e.g., self-distillation, the generation of self-distillation instances, and diversity-aware reinforcement learning), is thoughtfully designed and empirically validated through comprehensive ablation studies. These techniques might influence future research in this area.

**Weaknesses:**

- The overall writing could be improved. Some parts of the paper are difficult to follow at first reading. For example, the description related to Figure 3(c) is somewhat confusing, as the detailed definitions and procedures are only provided in the appendix. It would improve readability if part of this information were moved to the main text to make the presentation more self-contained.

- In Algorithm 1, some referenced functions are not rigorously defined (e.g., Verify($\hat{Y}, Y*$)). While their meanings can be intuitively understood, a more formal and precise specification would improve clarity. In addition, $\hat{Y}$ and $Y*$ actually represent different entities: $\hat{Y}$ denotes a reasoning trajectory, whereas $Y*$ corresponds only to the ground-truth tool-use information. The current notation may lead readers to mistakenly assume they are of the same type. It would be clearer to use distinct notation (for example, a $\mathcal{\hat{Y}}$ for the trajectory) to better reflect this distinction.

- The potential iterative application of the proposed framework is not discussed in the paper. Since the combination of self-distillation and reinforcement learning could be executed repeatedly, it would be valuable to analyze whether iterative cycles of D-CORE would lead to further improvements or possible instability. In addition, the paper does not discuss a simple yet relevant baseline: explicitly prompting the model to reduce Lazy Reasoning during reasoning. Including such an analysis would help clarify how much of the improvement comes from architectural innovation versus prompt-level guidance.

**Questions:**

- I noticed that the algorithm may return an empty set in some cases (e.g., line 2 of Algorithm 1). Does this mean that if the LLM performs an incorrect decomposition, the corresponding sample is discarded, and a new one must be generated until a valid trajectory is obtained? Also, is the success rate of this decomposition process the one reported in Table 7? If so, how does this success rate change after SFT/RL? Does the distilled model show a consistent improvement in decomposition success rate compared to the initial one?

---

> ### Author Response · Authors · 2025-11-22
> **Response to Reviewer q2LN (1/3)**
>
> We thank Reviewer `q2LN` for acknowledging our work and for raising very valuable questions. We will address your questions one by one and hope to resolve your concerns.
>
> > **Q1: if the LLM performs an incorrect decomposition, the corresponding sample is discarded, and a new one must be generated until a valid trajectory is obtained? Also, is the success rate of this decomposition process the one reported in Table 7?**
>
> A1: Your understanding is correct. In our approach, samples with incorrect decompositions are indeed discarded. The decomposition success rate is reported in Table 7, which indicates that Algorithm 1 results in approximately 7% data loss. We spent significant time refining this task decomposition prompt scheme. Eventually, we determined that a 93.2% success rate was adequate for generating the vast majority of high-quality self-distillation data. In addition, we also tried an immediate retry scheme upon failure, which slightly improved the success rate.
>
> | Model | Success Rate(%) |
> |-------|:---------------------------:|
> | Qwen3-8B | 93.2 |
> | Retry | 93.8 |
>
> >**Q2: how does this success rate change after SFT/RL? Does the distilled model show a consistent improvement in decomposition success rate compared to the initial one?**
>
> A2: Thank you for this important concern. To address your question, we conducted comparative experiments evaluating the task decomposition success rates using models after self-distillation and after RL training. The results are presented below:
>
> | Model | Success Rate(%) |
> |-------|:---------------------------:|
> | Qwen3-8B | 93.2 |
> | Self-distillation | 91.0 |
> | D-CORE-8B | 91.2 |
>
> As the results demonstrate, the decomposition success rates of these two models show no improvement and even exhibit a slight decline. The reason for this phenomenon is straightforward: our training objective is designed to enable the model to learn, during the reasoning process, how to recognize tasks, decompose them, and reason about executing the current subtask. Importantly, we do not explicitly optimize the model's pure task decomposition capability as an isolated skill.
>
> This design choice reflects our focus on end-to-end reasoning performance rather than task decomposition as a standalone metric. We appreciate your raising this point, as it allows us to clarify the specific scope and objectives of our training methodology. We are committed to addressing any concerns regarding the practical implications of this approach and welcome further discussion on this matter.

---

> ### Author Response · Authors · 2025-11-22
> **Response to Reviewer q2LN (2/3)**
>
> >**W1: analyze whether iterative cycles of D-CORE would lead to further improvements or possible instability.**
>
> A1: Thank you for this insightful suggestion regarding iterative refinement.  We have performed two complete additional iterations of the D-CORE algorithm. The results on BFCLv3 and $\tau$-Bench indicate that additional iterations neither enhance performance nor cause training collapse, suggesting that our method has reached a stable convergence point:
>
> | Model | BFCLv3 Overall | $\tau$-Bench Overall |
> |-------|:----------------:|:-------------------:|
> | Qwen3-8B | 66.3 | 29.0 |
> | D-CORE-iteration1 | 77.7 | 47.6 |
> | D-CORE-iteration2 | 76.9 | 46.5 |
> | D-CORE-iteration3 | 76.6 | 47.1 |
>
> We deeply appreciate your thoughtful engagement with our work, we would like to share our analysis and insights on implementing iterative mechanisms in D-CORE：
>
> 1.You correctly identify an important distinction between D-CORE and iterative methods like SELF-REFINE [1]. Our current approach implements only the *refine* component (task decomposition based on ground truth), but lacks the *feedback* mechanism that drives extra iterations of self-improvement loops.
>
> 2.Our analysis reveals that the current tool-use datasets are not sufficiently challenging for Qwen3-8B/14B, as evidenced by the high decomposition success rates (93.2%) when using GT-based supervision. This suggests that the primary limitation is not decomposition capability, but rather the Lazy Reasoning phenomenon we identified. However, this insight points to a promising direction: **on tasks where success rates remain low even after eliminating Lazy Reasoning**—such as BFCLv4-agentic (where D-CORE-8B/14B show less pronounced improvements), tasks involving agent memory, or search engine integration—we expect decomposition success rates to decrease substantially. In such scenarios, an iterative D-CORE approach would likely outperform single-pass decomposition significantly.
>
> 3.Building on our supplementary experiments for Reviewer dxrp, we demonstrated that pseudo-labels from capable open-source models can achieve ~90% of GT effectiveness. Combining this with the insights above, we envision a practical iterative framework:
>
>  1. Collect complex agentic tasks (e.g., multi-step planning, memory utilization)
>
> 2. Generate pseudo-labels $y$ using strong open-source models
>
> 3. Apply modified Algorithm 1 in paper :
> Input: ($C$, $Q$, $y$)
> Output: Both $\hat{y}$ and decomposed steps $S$
>
> 4. Multi-objective SFT:
> Learn ($C$, $Q$) → $\hat{y}$ (end-to-end reasoning)
> Learn $Q$ → $S$(decomposition capability)
>
> 5. Iterate with increasingly complex tasks.
>
> [1] Madaan A, Tandon N, Gupta P, et al. Self-refine: Iterative refinement with self-feedback[J]. Advances in Neural Information Processing Systems, 2023, 36: 46534-46594.

---

> ### Author Response · Authors · 2025-11-22
> **Response to Reviewer q2LN (3/3)**
>
> > **W2: Some parts of the paper are difficult to follow at first reading. the description related to Figure 3(c) is somewhat confusing, as the detailed definitions and procedures are only provided in the appendix.**
>
> A2: Thank you for this valuable suggestion. We have revised this section in the main text by moving the experimental setup details from the appendix into the main body. This revision makes it easier for readers to understand how the results in Figure 3(c) were obtained. The revised content is highlighted in blue in the newly uploaded PDF. We sincerely appreciate your constructive feedback.
>
> > **W3: In Algorithm 1, some referenced functions are not rigorously defined,  $\hat
> {Y}$and$Y^*$actually represent different entities，$\mathcal{\hat{y}}$ for the trajectory to better reflect this distinction.**
>
> A3: Thank you for pointing this out. We have corrected the notation for $\hat{Y}$ in Algorithm 1 according to your suggestion. We sincerely appreciate you taking the time to carefully read through our methodology and algorithm sections and for providing such valuable suggestions. Thank you for your thorough engagement with our work.
>
> > **W4: the paper does not discuss a simple yet relevant baseline: explicitly prompting the model to reduce Lazy Reasoning during reasoning.**
>
> A4: We conducted a controlled experiment on Qwen3-8B/14B with explicit prompting to reduce Lazy Reasoning. The experimental conditions were the same as those described in our appendix, where we defined Lazy Reasoning as reasoning processes exceeding 300 tokens in length or containing more than 3 reflection tokens, and dataset is BFCLv3 Multi-turn Base.
>
> | Model | Lazy Reasoning Ratio (%)| Accuracy (%)|
> |:-------|:---------------------:|:----------:|
> | Qwen3-8B | 42.7 | 38.0 |
> | Decomposed Prompt | 6.7 | 49.5 |
> | D-CORE-8B | 3.4 | 75.0 |
>
> | Model | Lazy Reasoning Ratio (%)| Accuracy (%)|
> |:-------|:---------------------:|:----------:|
> | Qwen3-14B | 40.8 | 48.0 |
> | Decomposed Prompt | 8.5 | 60.0 |
> | D-CORE-14B | 3.5 | 78.0 |
>
> The experimental results demonstrate that using decomposed prompting can significantly reduce the ratio of Lazy Reasoning while improving the model's performance in multi-turn scenarios. Furthermore, after applying our D-CORE approach, both the Lazy Reasoning ratio and Accuracy achieved further improvements, which fully validates the effectiveness of our proposed method.
>
> We sincerely thank the reviewer `q2LN` for the constructive feedback. We have carefully addressed each concern through additional experiments, enhanced analysis, and comprehensive revisions. We are committed to further refinements if any concerns remain.

---

> > ### Comment · Reviewer_q2LN · 2025-11-26
> >
> > Thanks for the clarifications provided in the rebuttal. I appreciate the authors’ efforts in addressing the concerns raised and presenting additional experimental results. I have no further comments at this stage.

---

> > > ### Author Response · Authors · 2025-11-26
> > > **Thank you for your positive feedback**
> > >
> > > We sincerely thank the reviewer for the positive feedback and for confirming that our rebuttal has addressed the concerns raised. We greatly appreciate the constructive comments throughout the review process, which have substantially improved the quality of our paper. We are grateful for the reviewer's time and consideration.

---

### Official Review · Reviewer_dxrp · 2025-11-01

**Soundness:** 3
**Presentation:** 3
**Contribution:** 2
**Rating:** 4
**Confidence:** 3

**Summary:**

This paper introduces D-CORE, a two stage training framework designed to enhance the complex tool use capabilities of Large Reasoning Models (LRMs). The authors identify a recurring failure mode termed “Lazy Reasoning”, in which models engage in repetitive, low value reflective reasoning rather than decomposing complex tasks effectively.

D-CORE addresses this by first employing self-distillation to teach structured task decomposition, followed by diversity-aware reinforcement learning (DA-GRPO) to restore reflective reasoning diversity. The method achieves strong empirical results, notably 79.3% accuracy on BFCLv3 and state-of-the-art performance on τ-bench, with models up to five times smaller outperforming larger baselines. The work presents an organized empirical study, clear problem framing, and impactful results on benchmarks of practical importance.

The paper is well motivated, clearly presented, and empirically strong, offering valuable insights into the limitations of reasoning in large models and proposing a structured remedy with convincing results. However, the core methodological innovation feels incremental, relying heavily on ground truth assisted data generation and established reinforcement learning heuristics. The contribution is thus more of a careful systemization and validation of known ideas rather than a breakthrough in reasoning architectures.

In summary, this is a solid and impactful piece of empirical work with clear value for the community, but its conceptual depth and theoretical originality fall slightly short of the standard expected for acceptance. With additional evidence of generalization and stronger justification of its mechanisms, this work could become a meaningful contribution in subsequent revisions.

**Strengths:**

Clear Problem Definition and Motivation:
The paper compellingly identifies and analyzes the “Lazy Reasoning” phenomenon in LRMs, providing behavioral and quantitative evidence of how excessive reflection impedes performance in multi-turn tool-use scenarios. This focus on reasoning efficiency rather than sheer model size is a valuable direction for scaling reasoning systems.

Strong Empirical Validation:
Results are robust and consistent across multiple model sizes and benchmarks. The reported +30% improvement in multi-turn performance and significant gains over ToolRL and Qwen baselines demonstrate that D-CORE achieves real, measurable advances. The use of ablations to isolate the effects of self-distillation and DA-GRPO is systematic and convincing.

Methodical and Transparent Experimental Design:
The paper offers well-structured experiments and diagnostic analyses. The manual decomposition experiment (Fig. 4b) substantiates the hypothesis that explicit decomposition mitigates lazy reasoning. The inclusion of detailed algorithms, prompt templates, and multi-model evaluations enhances credibility and reproducibility.

Clarity and Presentation:
The writing is clear and logically structured, supported by intuitive figures (especially Figures 1 and 5) that convey the framework and reasoning behavior differences. The presentation of results and ablations is thorough without being overwhelming.

Practical Relevance:
The efficiency gains, where a 14B model matches or surpasses 70B baselines, underscore the significance of the contribution for real world applications in reasoning centric AI systems and agentic tool use.

**Weaknesses:**

Limited Novelty of Core Mechanisms:
The core technical ideas, self distillation using ground truth guided decomposition and entropy-based reinforcement learning, are adaptations of existing concepts rather than fundamental innovations. Stage 1 primarily represents a structured data-generation and fine tuning pipeline, while Stage 2 employs a standard entropy based exploration adjustment. The methodology feels more engineering-driven than theoretically novel.

Dependence on Ground Truth for Decomposition:
A key limitation lies in the reliance on access to ground truth trajectories during the self distillation phase. As shown in Table 7, decomposition success rates drop from 93% to 73% without this supervision. This dependence reduces the framework’s applicability to domains where labeled trajectories are unavailable.

Fragile Reinforcement Stage:
The DA-GRPO component assumes that higher entropy equates to productive reflection, but this connection is empirically delicate. The paper itself notes that overly large entropy coefficients reintroduce unproductive “lazy reasoning,” suggesting that the framework may require careful, task specific tuning to remain stable.

Incomplete Theoretical Justification:
The reasoning behind the entropy advantage link is intuitive but not analytically supported. The framework would benefit from deeper theoretical grounding or stronger empirical validation showing that entropy directly correlates with meaningful reflection rather than randomness.

Limited Generalization Analysis:
Although results on BFCLv3 and τ-bench are strong, it remains unclear how D-CORE performs in out-of-distribution or real world settings where task structures and tools differ significantly. Broader validation or zero shot experiments would strengthen claims of generality.

**Questions:**

Ground Truth Dependence: How does D-CORE perform in data-sparse or unsupervised settings where ground-truth outputs are unavailable for decomposition and verification?

Entropy–Reflection Relationship: Can the authors empirically demonstrate that the entropy-based advantage term promotes useful reflection rather than random exploration?

Unified Learning Framework: Could a single training objective jointly encourage decomposition and reflection, eliminating the need for a sequential “break-and-fix” two-stage setup?

Generalization: Have the authors tested D-CORE on unseen tool-use domains or tasks beyond BFCLv3 and τ-bench to assess adaptability?

Failure Handling: When decomposition fails (as indicated by the 26–30% failure rate without ground truth), are such cases discarded, or do they contribute to training in some corrective form?

---

> ### Author Response · Authors · 2025-11-22
> **Response to Reviewer dxrp(1/4)**
>
> We thank Reviewer `dxrp` for the positive feedback and valuable questions. We address each question below and hope to clarify your concerns.
> >**Q1: Generalization: Have the authors tested D-CORE on unseen tool-use domains or tasks beyond BFCLv3 and τ-bench to assess adaptability?**
>
> A1:Since you and Reviewer`jzW5` share the same concern, we have prepared a unified response to address both of your questions. To thoroughly address your concerns about generalization, we have conducted three additional sets of experiments on $\tau^2$-Bench, ACEBench, and BFCLv4-Agentic :
>
> -  $\tau^2$-Bench is a novel benchmark for evaluating conversational agents in dual-control environments, where both the user and the agent can perform actions to affect the shared world state.  Our experimental results demonstrate that D-CORE maintains strong performance even under these more demanding conditions, particularly on the newly introduced telecom tasks, which validates the effectiveness of our approach.
> | Model | Retail &uarr;| Airline &uarr;| Telecom &uarr;| Avg&uarr; |
> |:------|:-------|:--------|:--------|:----|
> | Qwen3-8B | 41.5 | 31.3 | 26.3 | 33.0 |
> | Qwen3-14B | 46.5 | 30.0 | 31.7 | 36.1 |
> | Qwen3-32B | 49.1 | 36.0 | 28.4 | 37.8 |
> | xLAM2-8B | 51.3 | 35.4 | 22.4 | 36.4 |
> | xLAM2-32B | 53.1 | 41.6 | 26.1 | 40.3 |
> | xLAM2-70B | **54.9** | **46.0** | 29.3 | 43.4 |
> | D-CORE-8B | 49.8 | 38.4 | 30.5 | 39.6 |
> | D-CORE-14B | 53.5 | 44.1 | **34.9** | **44.2** |
> - ACEBench employs more complex system and user prompts, revealing that fine-tuned LLMs that perform well on BFCL actually underperform compared to open-source general-purpose models on this benchmark. This confirms that SFT-trained LLMs suffer from degraded generalization capabilities. The experimental results on ACEBench demonstrate that our model achieves further improvements over the Qwen3 baseline.
> | Model | Normal&uarr; | Special&uarr; | Agent &uarr;| Overall &uarr;|
> |:------|:------:|:-------:|:-----:|:-------:|
> | Qwen3-8B | 71.4 | 75.3 | 29.1 | 65.9 |
> | Qwen3-14B | 66.9 | 84.0 | 44.2 | 68.0 |
> | Qwen3-32B | 75.9 | 77.3 | 49.2 | 72.2 |
> | xLAM2-8B | 58.8 | 0.0 | 5.0 | 34.8 |
> | xLAM2-32B | 69.2 | 24.7 | 13.4 | 52.5 |
> | xLAM2-70B | 57.1 | 5.3 | 38.4 | 36.5 |
> | D-CORE-8B | **77.9** | 78.7 | **59.2** | 75.2 |
> | D-CORE-14B | 77.1 | **88.7** | 55.8 | **76.9** |
>
> - BFCLv4-agentic introduces two innovative evaluation scenarios: 1.Web-search task: Tests multi-hop question answering capabilities using web search tools. 2.Memory task: Evaluates conversational memory management using dedicated memory tools.
> | Model | Memory KV&uarr; |Memory Vector&uarr;  |Memory  Recursive Sum&uarr; |  Web Search Base &uarr;|
> |-------|---------------|-------------------|-------------|---------------|
> | Qwen3-8B | 14.8 | 14.8 | 27.1 | 16.0 |
> | Qwen3-14B | 15.5 | 17.4 | 42.6 | 34.0 |
> | Qwen3-32B | **15.5** | 20.7 | 38.7 | 32.0 |
> | xLAM2-8B | 4.5 | 14.2 | 28.4 | 8.0 |
> | xLAM2-32B | 8.4 | 10.4 | 31.6 | 31.0 |
> | xLAM2-70B | 5.2 | 11.6 | 36.1 | 13.0 |
> | D-CORE-8B | 14.0 | 17.0 | 23.0 | 36.0 |
> | D-CORE-14B | 11.6 | **24.9** | **42.9** | **39.0** |
>
> D-CORE achieves substantial improvements on the web-search task while maintaining baseline performance on the memory task.  After conducting a thorough analysis, we identified a strong correlation between the multi-hop nature of web search queries and our task decomposition approach:
> ```
> user: "Many of the world's largest waterfalls are surrounded by vast jungles that contribute to their ecological richness. Which jungle surrounds the national park that has the largest waterfall in the country with the highest GDP growth in 2024?"
> ```
> ```
> D-CORE-8B: <think> To complete this task, I need to break it down into the following subtasks:
>
> 1. Find the country with the highest GDP growth in 2024.
>
> 2. Find the largest waterfall in that country.
>
> 3. Find the national park that contains this waterfall.
>
> 4. Identify what jungle surrounds this national park.
>
> Now I should analyze the execution process of subtask 1. Find the country with the highest GDP growth in 2024. Let me start by searching for the country with the highest GDP growth in 2024. <\think> [search_engine_query(keywords='country highest GDP growth 2024')]
> ```
> These experimental results demonstrate that the decomposition capabilities learned from complex instructions successfully transfer to the new domain, providing valuable insights for our future work.
>
> We sincerely thank you for raising this important question. We hope our experimental findings address your concerns. Your feedback has genuinely enhanced the quality and scope of our research.

---

> ### Author Response · Authors · 2025-11-22
> **Response to Reviewer dxrp (2/4)**
>
> > **Q2: Entropy–Reflection Relationship: Can the authors empirically demonstrate that the entropy-based advantage term promotes useful reflection rather than random exploration?**
>
> A2:Thank you for this insightful question. We have created detailed plots showing the evolution of reward, reflection token numbers, and entropy during DA-GRPO training on tool use dataset. These plots are included in the revised paper as Figure 7. The figure clearly shows that adding entropy-based advantages yields three effects: higher reward, more reflection tokens, and lower entropy. These trends match [1], confirming DA-GRPO's role in shaping RL training dynamics.
>
> [1] Cheng D, Huang S, Zhu X, et al. Reasoning with exploration: An entropy perspective[J]. arXiv preprint arXiv:2506.14758, 2025.
>
> >**Q3:Ground Truth Dependence: How does D-CORE perform in data-sparse or unsupervised settings where ground-truth outputs are unavailable for decomposition and verification?**
>
> A3:Dear Reviewer, Thank you for this excellent question. To systematically evaluate the effectiveness of our method under sparse or missing ground-truth supervision, we conducted a rigorous set of ablation experiments. We designed three experimental conditions using APIGen-MT-5k [1] training samples and Qwen3-8B as the base model:
> 1. **GT**: We perform standard D-CORE on Qwen3-8B using the 5,000 samples with full ground-truth labels.
> 2. **Pseudo Label**: We use Qwen3-Max to generate pseudo-labels for the 5,000 samples, then apply the complete D-CORE pipeline using these pseudo-labels for self-distillation.
> 3. **Unsupervised**: We execute the D-CORE pipeline without any label—neither ground truth nor pseudo-labels—relying entirely on the model's self-generated decompositions.
>
> We evaluate all three conditions on BFCL and $\tau$-Bench. The results are presented below:
> | Model | BFCL Live &uarr;| BFCL Irrelevance &uarr;| BFCL Multi-Turn &uarr;| BFCL  non-live &uarr;| BFCL Overall &uarr;| $\tau$-bench Retail &uarr;| $\tau$-bench Airline &uarr; | $\tau$-bench Overall &uarr;|
> |:---|:---:|:---:|:---:|:---:|:---:|:---:|:---:|:---:|
> | Qwen3-8B | 78.5 | 79.1 | 33.0 | 88.8 | 64.0 | 34.7 | 23.2 | 29.0 |
> | GT | 79.7 | 86.2 | 41.6 | **83.1** | 68.4 | **43.8** | **30.4** | **37.1** |
> | Pseudo Label | **81.2** | **95.1** | **46.1** | 72.7 | **68.5** | 38.5 | 20.7 | 29.6 |
> | Unsupervised | 80.0 | 87.2 | 40.4 | 83.6 | 68.4 | 18.6 | 22.0 | 20.3 |
>
> As shown in the table, D-CORE achieves limited improvement in the unsupervised setting, primarily due to the low accuracy of task decomposition. However, we are particularly encouraged by the pseudo label results, where D-CORE achieves 90% of the performance obtained with ground truth labels. This experiment reveals a promising finding: in scenarios with sparse ground truth annotations, we can leverage low-cost pseudo labeling as a reliable approach to establish a solid performance lower bound for the D-CORE method.
> Thank you again for this insightful question.
>
> [1]https://huggingface.co/datasets/Salesforce/APIGen-MT-5k
>
> > **Q4: Failure Handling: When decomposition fails (as indicated by the 26–30% failure rate without ground truth), are such cases discarded, or do they contribute to training in some corrective form?**
>
> A4: In our approach, samples with incorrect decompositions are indeed discarded. The decomposition success rate is reported in Table 7, which indicates that Algorithm 1 results in approximately 7% data loss. We spent significant time refining this task decomposition prompt scheme. Eventually, we determined that a 93.2% success rate was adequate for generating the vast majority of high-quality self-distillation data. In addition, we also tried an immediate retry scheme upon failure, which slightly improved the success rate.
> | Model | Success Rate(%) |
> |-------|:-------------------:|
> | Qwen3-8B | 93.2 |
> | Retry | 93.8 |
>
> Additionally, we fully appreciate the reviewer's concern about the absence of ground truth. To directly address this issue, we conducted a supplementary experiment in which we employed the Qwen3-Max model to generate pseudo-labels in place of ground truth, thereby simulating scenarios where GT is entirely unavailable. Our results demonstrate that Algorithm 1 maintains a strong decomposition success rate of 92.8% even under these conditions.
>
> | Reference Trajectories | Success Rate(%) |
> |-------|:-------------------:|
> |Ground Truth | 93.2 |
> |Pseudo Labels | 92.8 |
>
> This experimental result establishes a lower bound for our method, ensuring it does not degrade to the 26–30% failure rate observed in baseline approaches.

---

> ### Author Response · Authors · 2025-11-22
> **Resonse to Reviewer dxrp (3/4)**
>
> > **Q5:Unified Learning Framework: Could a single training objective jointly encourage decomposition and reflection, eliminating the need for a sequential “break-and-fix” two-stage setup?**
>
> A5: Thank you for this insightful question about unified learning frameworks.
> We agree that joint optimization is theoretically appealing. However, after
> careful empirical investigation, we adopted the two-stage approach for the
> following reasons:
>
> 1. **Empirical evidence strongly supports decomposition-first learning:**
>    - Recent work [1] demonstrated that outcome-based RLVR does not elicit new
>      reasoning patterns—we directly validated this in Figure 3(d) and Table 5 in our paper, where RL-only
>      models achieve 6.2% lower than base model on BFCLv3 MT.
>    - Conversely, studies [1,2,3] show distillation successfully transfers complex
>      reasoning patterns, which our ablations confirm (Table 5: SFT stage improves
>      accuracy by 24.5% and 7.6% on BFCLv3 MT and $\tau$-bench).
>    - Table 5 also clearly demonstrates that applying GRPO after SFT yields further substantial improvements, with gains of 9.9 % on BFCLv3 MT and 8.6 % on $\tau$-bench."
>
> 2. **Process-reward RL requires further investigation:**
>     -  As detailed in the Appendix, our two-stage method for Qwen3-8B required approximately: 30 hours for self-distillation data generation, 11 hours for SFT, 11 hours for RL, Total: ~52 hours.
>     - We conducted a preliminary cost-benefit analysis for process-reward RL using LLM-as-judge. Online RL with LLM-as-judge for equivalent data scale would require approximately 200 hours per experiment.
>     - Given our resource constraints and the need to conduct multiple ablations, this significantly impacted our ability to deliver rigorous empirical results within a reasonable timeframe.
>
>
> We are genuinely grateful that you invested the time to thoroughly read and understand our paper—your question demonstrates a deep engagement with our work. Thank you once again for your constructive feedback and recognition of our contributions.
>
> [1] Yang Yue, Zhiqi Chen, Rui Lu, Andrew Zhao, Zhaokai Wang, Shiji Song, and Gao Huang. Does reinforcement learning really incentivize reasoning capacity in llms beyond the base model? arXiv
> preprint arXiv:2504.13837, 2025
>
> [2] Tianzhe Chu, Yuexiang Zhai, Jihan Yang, Shengbang Tong, Saining Xie, Dale Schuurmans, Quoc Le, Sergey Levine, and Yi Ma. Sft memorizes, rl generalizes: A comparative study of foundation
> model post-training. arXiv preprint arXiv:2501.17161, 2025
>
> [3] Guo D, Yang D, Zhang H, et al. Deepseek-r1: Incentivizing reasoning capability in llms via reinforcement learning[J]. arXiv preprint arXiv:2501.12948, 2025.

---

> ### Author Response · Authors · 2025-11-26
> **Supplementary Experiments on DA-GRPO**
>
> Dear Reviewer `dxrp`,
> to further address your concerns regarding DA-GRPO, we have conducted an additional experiment to validate its effectiveness.
>
> In the revised manuscript, we have supplemented the motivation for DA-GRPO. The advantage $A_{i,t}$ in GRPO is defined as: $A_{i,t} = \frac{R_i - mean({R_i})}{std({R_i})}$. When the standard deviation $std({R_i})$ approaches zero, the advantage term $A_{i,t}$ vanishes, thereby eliminating the gradient signal.  This motivates us to seek an alternative formulation. Recent work [1][2] shows that high-entropy tokens correlate with reflective reasoning, which we confirm in Figure 6(b), and that entropy-based advantages promote reasoning diversity [1]. This leads to our motivation: **can we leverage an entropy-based advantage to address gradient vanishing when std approaches zero, thus recovering reasoning diversity?**
>
> The key idea is to scale each token’s gradient update in proportion to its detached entropy $\mathcal{H}^{\mathrm{detach}}$ when std approaches zero:
>
> $\mathcal{H}^{\mathrm{detach}}=-\sum_{v\in V}\pi_\theta(v|q,o_{<t})\log\pi_\theta(v|q,o_{<t}).$
>
> The entropy term $\mathcal{H}^{\mathrm{detach}}_{i,t}$ is detached during backpropogation, serving as a fixed offset that adjust update magnitude without affecting gradients. We then define an entropy-based advantage term $\psi(\mathcal{H}_t)$ to incentivize the generation of high-entropy tokens:
>
> $\psi(\mathcal{H}_t)=\min({\alpha} \cdot {{H}^{{detach}}},~\delta)$.
>
> Then we define the advantage function in DA-GRPO as:
>
> $\hat{A}_{i,t} = \psi (\mathcal{H_t})$ when $std({R_i})=0$, otherwise, it equals the original advantage.
>
> To validate the effectiveness of DA-GRPO, we conducted experiments using the dynamic sampling method proposed in DAPO [3]. Specifically, we over-sample and filter out prompts with $std({R_i})=0$, retaining only prompts with effective gradients in each batch while maintaining a consistent batch size. The results show that simply discarding samples with zero standard deviation yields suboptimal performance. For evaluation, we selected $\tau$-bench, where DA-GRPO achieves the most substantial improvements, along with the out-of-distribution $\tau^2$-bench dataset, to clearly demonstrate the efficacy of DA-GRPO:
>
> | Method || $\tau$-bench | || $\tau^2$-bench ||| Avg |
> |:--------:|:---------:|:---------:|:---------:|:--------:|:-------:|:-------:|:-------:|:--------:|
> | | Retail &uarr;| Airline &uarr;| Overall &uarr;| Retail &uarr;| Airline &uarr;| Telecom &uarr;| Overall &uarr;| (%)&uarr; |
> | GRPO | 49.2 | 41.2 | 45.2 | **50.2** | **40.1** | **34.0** | **41.4** | 43.3 |
> | GRPO + Dynamic Sampling | 56.8 | 38.8 | 47.8 | 49.5 | 35.3 | 27.0 | 37.3 | 42.6 |
> | DA-GRPO α=0.1 | **59.8** | **44.8** | **52.3** | 49.8 | 38.4 | 30.5 | 39.6 | **45.9** |
>
> The experimental results confirm that while dynamic sampling improves performance on $\tau$-bench by discarding samples with zero standard deviation, it degrades performance on the out-of-distribution $\tau^2$-bench. In contrast, DA-GRPO, which computes token entropy advantages for samples with zero standard deviation, achieves the most balanced performance across both benchmarks.
>
> We hope this experiment alleviates your concerns about DA-GRPO and welcome your feedback.
>
> [1] Cheng, Daixuan, et al. "Reasoning with exploration: An entropy perspective." arXiv preprint arXiv:2506.14758 (2025).
>
> [2] Wang, Shenzhi, et al. "Beyond the 80/20 rule: High-entropy minority tokens drive effective reinforcement learning for llm reasoning." arXiv preprint arXiv:2506.01939 (2025).
>
> [3] Yu, Qiying, et al. "Dapo: An open-source llm reinforcement learning system at scale." arXiv preprint arXiv:2503.14476 (2025).

---

> ### Author Response · Authors · 2025-11-27
> **Seeking Feedback on Our Rebuttal Responses**
>
> Dear Reviewer `dxrp`,
>
> Thank you for your initial review. We have provided a detailed rebuttal addressing all the concerns raised in your review.
>
> We notice that some of the points mentioned may benefit from further clarification based on our responses. We would greatly appreciate it if you could take a moment to review our rebuttal and provide feedback on whether our clarifications adequately address your concerns.
>
> We are committed to improving our work and value your expert guidance. If any aspects remain unclear or require additional experiments, we are happy to provide further information during the discussion period.
>
> Looking forward to your response.
>
> Best regards,
> Authors

---

> ### Author Response · Authors · 2025-12-02
> **Response to Reviewer dxrp (4/4)**
>
> > **W1:Limited Novelty of Core Mechanisms: ... The methodology feels more engineering-driven than theoretically novel.**
>
> A1: We appreciate the reviewer’s comment. While our approach thoughtfully integrates established concepts (distillation, entropy-based RL), our primary contribution is not in proposing a new optimization primitive, but rather in identifying and formalizing a novel and chanllenging problem setting: **complex tool use for LRMs**. We subsequantly introduce a training paradigm to address the unique complexities of this setting, which, to our knowledge, represents a gap in prior work.
>
> 1.Novel Problem Setting
>
> Most existing work on tool use LRMs focuses either on single-turn scenarios, or distilling from stronger teacher models that already exhibit robust reasoning. We explicitly study a different setting: **Given an LRM that already performs strongly on single-turn tool use, can we fix its multi-turn Lazy Reasoning without any external teacher?** Section 2 provides two key empirical findings that define this setting:
>
> - **LRMs have substantial untapped tool-use potential.** Figure 2 shows that, on non-multi-turn tool use tasks with explicit reasoning, LRMs can significantly outperform LLMs. This indicates that the necessary capabilities for complex tool use are already present in the model.
>
> - **Lazy reasoning is primarily an elicitation failure.** In complex multi-turn conversations,  LRM exhibits “Lazy Reasoning” . However, Figure 4 demonstrates that simple decomposed prompting almost entirely removes this behavior. This suggests the model possesses latent tool use skills that are not properly elicited by standard prompts.
>
> Our methods are designed specifically for this teacher-free elicitation and consolidation problem, which we believe is itself a meaningful novelty.
>
> 2.Self-Distillation
>
> We agree that “data generation + fine-tuning” is standard. The novelty here is how we instantiate it for LRMs in this setting:
>
> - **Pattern-aware decomposition for tool use.** We introduce formal definitions of sequential, parallel, and irrelevant tool calls in tool-use tasks, and propose pattern-specific strategies for decomposition and reasoning composition. This structured analysis of tool use patterns has not been explored in existing self-distillation or self-improvement literature[4,5,6].
>
> - **Teacher-free self-improvement for LRMs.** Existing LRM research relies exclusively on distilling reasoning from larger teacher models. We demonstrate that high-quality decomposed reasoning can be self-generated when properly guided.
>
> After self-distillation, we observe:
>
> - increased decomposition thoughts and reduced Lazy Reasoning (Figure 8),
>
> - substantial gains on BFCLv3 and τ-bench (Tables 1–2),
>
> - robust generalization to out-of-distribution tasks (Table 4).
>
> These results indicate that we are not just learn the reasoning format, but incentivizing a general task decomposition reasoning to complex tool use.
>
> 3.DA-GRPO:
>
> We do not claim entropy itself is new. DA-GRPO is a targeted modification of GRPO designed for a specific feature created by self-distillation.  Self-distillation increases decomposition frequency but reduces reflection (Figures 6, 8), causing many prompts to have zero reward standard deviation:
>
> - Standard GRPO cannot exploit these zero-variance trajectories.
>
> - Following DAPO [3], one can discard std=0 prompts via dynamic sampling, but this wastes data—especially problematic in multi-turn tool use, where trajectories are expensive.
>
> DA-GRPO addresses this low-variance reward regime: For normal-variance prompts, it behaves similarly to GRPO. For zero-variance prompts, it uses an entropy-based adjustment to still extract a useful learning signal, instead of discarding data.
>
> Empirically, DA-GRPO outperforms both GRPO and GRPO+dynamic sampling:
>
> | Method || $\tau$-bench | || $\tau^2$-bench ||| Avg |
> |:--------:|:---------:|:---------:|:---------:|:--------:|:-------:|:-------:|:-------:|:--------:|
> | | Retail | Airline | Overall | Retail | Airline | Telecom | Overall | (%) |
> | GRPO | 49.2 | 41.2 | 45.2 | **50.2** | **40.1** | **34.0** | **41.4** | 43.3 |
> | GRPO + Dynamic Sampling | 56.8 | 38.8 | 47.8 | 49.5 | 35.3 | 27.0 | 37.3 | 42.6 |
> | DA-GRPO α=0.1 | **59.8** | **44.8** | **52.3** | 49.8 | 38.4 | 30.5 | 39.6 | **45.9** |
>
> Thus, while DA-GRPO is not a new RL paradigm, it is a practical, variance-aware adaptation for the specific self-distilled regime we study.
>
> We hope this clarifies that our work goes beyond engineering a pipeline and offers a novel, well-motivated training paradigm and RL adaptation for multi-turn tool-using LRMs.
>
> References:
>
> [1] ToolRL: Reward is All Tool Learning Needs.
>
> [2] Nemotron-Research-Tool-N1.
>
> [3] Dapo: An open-source llm reinforcement learning system at scale.
>
> [4] Large language models can self-improve.
>
> [5] Toolformer: Language models can teach themselves to use tools.
>
> [6] self-refine: Iterative refinement with self-feedback.

---

### Author Response · Authors · 2025-11-22
**General Response**

Dear AC and Reviewers,

We sincerely thank all reviewers ( `dxrp`,  `q2LN`, `jzW5`) for their thoughtful feedback and constructive suggestions. We are encouraged that all reviewers recognized the significance of our contributions. All praised our identification of the "Lazy Reasoning" problem, novel D-CORE framework, and +30% performance gains. Reviewer `dxrp` particularly valued our efficiency gains (14B matching 70B) and transparency; Reviewer `q2LN` recognized our potential to shape future research; Reviewer `jzW5` appreciated our behavioral analysis for compositional tool-use reasoning.

Reviewers `dxrp` and `jzW5` both noted that our work would benefit from deeper theoretical grounding and stronger empirical validation. Specifically, Reviewer `dxrp` requested evidence that "entropy directly correlates with meaningful reflection rather than randomness," while Reviewer `jzW5` called for "deeper theoretical analysis, richer qualitative evidence, and clearer experimental documentation." We have substantially revised our manuscript to address these concerns.  All modifications are highlighted in **blue** and include the following changes:

1.We have corrected the grammatical issues pointed out by Reviewer `q2LN` and revised the description of Figure 3(c).

2.We have added the OOD experimental results and the analysis of DA-GRPO's training dynamics in Figure 7, as requested by Reviewers `dxrp` and `jzW5`.

3.Following Reviewer `jzW5`'s suggestion, we have included a qualitative analysis comparing the reasoning trajectories between Qwen3 and D-CORE in the Appendix.

4.In response to Reviewers `dxrp` and `jzW5`, we have expanded Section 3.2 with detailed motivation for DA-GRPO and added Figure 6 showing the impact of self-distillation on the model's std distribution as well as the distribution of high-entropy tokens.

We believe these revisions have significantly strengthened our paper and address the reviewers' concerns. Detailed responses to individual reviewers are provided below.

---

> ### Author Response · Authors · 2025-11-28
> **Revised Submission with Key Improvements**
>
> Dear Reviewers,
>
> Thank you for your valuable feedback. We have uploaded a substantially revised manuscript that directly addresses all of your concerns. For your convenience, all modifications are highlighted in blue:
>
> - **DA-GRPO Proof**: A rigorous mathematical proof is now included in the appendix.
>
> - **Section 3.2 Enhancement**: This section has been comprehensively rewritten to explicitly demonstrate how the entropy-based advantage mechanism enables GRPO to achieve effective learning even in cases where reward std=0.
>
> We respectfully ask that you review our revision. We believe the added theoretical rigor and enhanced clarity better demonstrate the practical value of our work. We hope these improvements address your concerns and meet the standards of ICLR.
>
> Thank you for your time and consideration.
>
> Sincerely,
> Authors

---

### Author Response · Authors · 2025-11-29
**Summary for new AC**

Dear AC，

I hope this message finds you well. First and foremost, please allow me to express our deep admiration and respect for you. Following the new ICLR policy announced on November 28th, we fully understand the incredible workload that each AC now bears.

Let me briefly summarize our paper's review situation, which is relatively straightforward. Our current scores are 8, 4, and 4, and have remained unchanged during the rebuttal period. We submitted all our rebuttal responses and updated our paper on November 22nd. By November 27th, reviewer `q2LN` (score: 8) responded to us, confirming that his/her concerns had been addressed. Reviewers `dxrp` (score: 4) and `jzW5` (score: 4) have not yet been able to engage during this period, which we fully understand given the intensive workload. Notably, both Reviewers `dxrp` and `jzW5` raised remarkably similar concerns. In response, we have prepared exceptionally detailed rebuttals for each reviewer, complete with new experiments, theoretical proofs, and comprehensive analyses that directly address every point raised.

Specifically,  Reviewer `dxrp` explicitly raised 5 questions and 1 key weakness. We appreciate these thoughtful comments, as they highlight areas where our initial presentation could be strengthened. In our rebuttal, we have systematically addressed each concern with substantial evidence: additional experiments (including OOD evaluation and training dynamics analysis), rigorous theoretical derivations (formal DA-GRPO proof in appendix and 3.2 in main text), and relevant citations supporting our design choices and novelty.

After reviewing our rebuttal, Reviewer `q2LN` responded: "***Thanks for the clarifications provided in the rebuttal. I appreciate the authors’ efforts in addressing the concerns raised and presenting additional experimental results. I have no further comments at this stage.***" We sincerely appreciate Reviewer `q2LN`'s constructive feedback and thoughtful engagement throughout the review process.

Reviewer `jzW5`  explicitly stated in the review: ***'...the paper would benefit from deeper theoretical analysis, richer qualitative evidence, and clearer experimental documentation...'*** In response, we have substantially strengthened our manuscript with rigorous mathematical proofs (Appendix), rich qualitative trajectory comparisons (Appendix), and detailed experimental analyses (Figures 6-7). We are confident these enhancements fully satisfy the reviewer's expectations.

We have substantially revised our manuscript to address the concerns raised,  with all modifications highlighted in **blue** for your convenience. We believe these revisions have significantly strengthened the paper and thoroughly addressed the reviewers' feedback. We sincerely appreciate your efforts in maintaining a fair and rigorous evaluation environment under challenging circumstances.

Sincerely,
Authors

---

### Meta-Review · Area_Chair_vQXc · 2025-12-29

**Summary:**

This paper presents D-CORE (Decomposing tasks and Composing Reasoning processes), a two-stage framework enhancing large reasoning models (LRMs) for complex multi-turn tool use. It identifies the "Lazy Reasoning" issue—LRMs’ unproductive lengthy reflections over effective task decomposition, and proposes to use self-distillation (for task decomposition/composition) and diversity-aware RL (DA-GRPO, restoring reflective exploration) to address it.

All three reviewers share common concerns: first, a weak theoretical foundation—the entropy advantage mechanism of DA-GRPO is only intuitively demonstrated, lacking formal correlation and convergence guarantees; second, insufficient generalization evidence, with initial evaluation limited to specific tool-use datasets; third, poor reproducibility, as key implementation details like dataset construction and prompt templates are not fully disclosed.

**Reviewer Concerns:**

Although the rebuttal has tried to address the aforementioned issues of the submission, the theoretical parts and reproducibility issue are still of concern.

**Reviewer Scores:**

Reviewer q2LN gives a rating of 8 (Accept), acknowledging the problem's value and framework innovation; both Reviewer dxrp and Reviewer jzW5 rate 4, focusing on theoretical depth, generalization, and reproducibility. The rebuttal partially alleviates concerns but fails to fully dispel key doubts, requiring further theoretical and detailed support.

---

### Decision · Program_Chairs · 2026-01-26

Reject